# Simple biochemical features underlie transcriptional activation domain diversity and dynamic, fuzzy binding to Mediator

Adrian L Sanborn[1,2]*, Benjamin T Yeh[2], Jordan T Feigerle[1], Cynthia V Hao[1], Raphael JL Townshend[2], Erez Lieberman Aiden[3,4], Ron O Dror[2], Roger D Kornberg[1]*

[1]Department of Structural Biology, Stanford University School of Medicine, Stanford, United States; [2]Department of Computer Science, Stanford University, Stanford, United States; [3]The Center for Genome Architecture, Baylor College of Medicine, Houston, United States; [4]Center for Theoretical Biological Physics, Rice University, Houston, United States

**Abstract** Gene activator proteins comprise distinct DNA-binding and transcriptional activation domains (ADs). Because few ADs have been described, we tested domains tiling all yeast transcription factors for activation in vivo and identified 150 ADs. By mRNA display, we showed that 73% of ADs bound the Med15 subunit of Mediator, and that binding strength was correlated with activation. AD-Mediator interaction in vitro was unaffected by a large excess of free activator protein, pointing to a dynamic mechanism of interaction. Structural modeling showed that ADs interact with Med15 without shape complementarity ('fuzzy' binding). ADs shared no sequence motifs, but mutagenesis revealed biochemical and structural constraints. Finally, a neural network trained on AD sequences accurately predicted ADs in human proteins and in other yeast proteins, including chromosomal proteins and chromatin remodeling complexes. These findings solve the longstanding enigma of AD structure and function and provide a rationale for their role in biology.

**\*For correspondence:**
a@adriansanborn.com (ALS);
kornberg@stanford.edu (RDK)

**Competing interests:** The authors declare that no competing interests exist.

## Introduction

Transcription factors (TFs) perform the last step in signal transduction pathways. They thus serve key roles in central processes such as growth, stress response, and development, and their mutation or misregulation underlies many human diseases (*Spitz and Furlong, 2012*). A TF includes a sequence-specific DNA-binding domain (DBD) and an effector domain that regulates nearby gene transcription. Activation domains (ADs) – effector domains that increase transcription – have long been of particular interest due to their roles as oncogenic drivers and use as scientific tools (*Bradner et al., 2017*; *Brückner et al., 2009*; *Dominguez et al., 2016*).

ADs were discovered as regions that could independently stimulate transcription when ectopically recruited to a gene promoter (*Brent and Ptashne, 1985*). Early experiments showed that ADs were unlike structured domains because progressive truncations showed graded reductions in activity (*Hope and Struhl, 1986*; *Hope et al., 1988*). Subsequent studies showed that ADs were disordered and had few similarities in their primary sequence (*Mitchell and Tjian, 1989*). Instead, ADs were classified based on their enrichment of certain residues, whether acidic, glutamine-rich, or proline-rich.

Acidic ADs are the most common and best characterized. Acidic ADs retain activity when transferred between yeast and animals, pointing to a conserved eukaryotic mechanism (*Fischer et al., 1988*; *Struhl, 1988*). While some have found that acidic residues are necessary for activation, others have found that they are dispensable (*Brzovic et al., 2011*; *Pacheco et al., 2018*; *Staller et al., 2018*; *Warfield et al., 2014*). Besides their negative charge, acidic ADs are rich in bulky hydrophobic

**eLife digest** Cells adapt and respond to changes by regulating the activity of their genes. To turn genes on or off, they use a family of proteins called transcription factors. Transcription factors influence specific but overlapping groups of genes, so that each gene is controlled by several transcription factors that act together like a dimmer switch to regulate gene activity.

The presence of transcription factors attracts proteins such as the Mediator complex, which activates genes by gathering the protein machines that read the genes. The more transcription factors are found near a specific gene, the more strongly they attract Mediator and the more active the gene is. A specific region on the transcription factor called the activation domain is necessary for this process. The biochemical sequences of these domains vary greatly between species, yet activation domains from, for example, yeast and human proteins are often interchangeable.

To understand why this is the case, Sanborn et al. analyzed the genome of baker's yeast and identified 150 activation domains, each very different in sequence. Three-quarters of them bound to a subunit of the Mediator complex called Med15. Sanborn et al. then developed a machine learning algorithm to predict activation domains in both yeast and humans. This algorithm also showed that negatively charged and greasy regions on the activation domains were essential to be activated by the Mediator complex.

Further analyses revealed that activation domains used different poses to bind multiple sites on Med15, a behavior known as 'fuzzy' binding. This creates a high overall affinity even though the binding strength at each individual site is low, enabling the protein complexes to remain dynamic. These weak interactions together permit fine control over the activity of several genes, allowing cells to respond quickly and precisely to many changes.

The computer algorithm used here provides a new way to identify activation domains across species and could improve our understanding of how living things grow, adapt and evolve. It could also give new insights into mechanisms of disease, particularly cancer, where transcription factors are often faulty.

residues. Mutating these hydrophobic residues reduces activation, often in proportion to the number mutated.

Because AD sequences are highly diverse and poorly conserved, only a small fraction of all ADs in eukaryotic TFs have likely been annotated. Sequence motifs have been proposed based on analysis of select ADs but have not been used for large-scale prediction (*Piskacek et al., 2007*). Screens of random sequences in yeast identified many activating sequences that represented as many as 1–4% of elements tested (*Hackett et al., 2020*; *Ravarani et al., 2018*). Actual protein sequences are, however, highly non-random. Direct screening of protein sequences has identified relatively few ADs at low resolution (*Arnold et al., 2018*; *Tycko et al., 2020*). There is a need for methods to experimentally detect or computationally predict all ADs.

ADs stimulate transcription of genes by recruiting coactivator complexes, especially the Mediator complex, which interacts with RNA Polymerase II and regulates its transcription initiation (*Kornberg, 2005*). Mediator is necessary for TF-dependent activation in vitro and in vivo and is required for regulation by enhancers in all eukaryotes (*Allen and Taatjes, 2015*). Mediator and TFs are concentrated at strong enhancers, perhaps in a phase-separated state, which may play a role in gene activation (*Boija et al., 2018*; *Chong et al., 2018*; *Sabari et al., 2018*; *Shrinivas et al., 2019*; *Whyte et al., 2013*). Beyond Mediator, TFs have been suggested to recruit other conserved multi-protein complexes, including TFIID, SAGA, and SWI/SNF, that play roles in activation of various genes (*Hahn and Young, 2011*; *Mitchell and Tjian, 1989*). While many such TF interactions have been described, their occurrence and roles remain to be determined.

Biochemical studies of TF-coactivator interaction have given insight into the mechanism of activation. Nuclear magnetic resonance (NMR) studies revealed short alpha helices formed by acidic ADs of yeast Gcn4 protein, with hydrophobic faces contacting hydrophobic surfaces of the Med15 subunit of Mediator (*Brzovic et al., 2011*; *Tuttle et al., 2018*). NMR constraints were consistent with multiple possible AD binding poses, suggesting a dynamic 'fuzzy complex' (*Tompa and Fuxreiter,*

*2008*). Further consistent with these ideas, solvent exposure of aromatic residues was associated with activation in a screen of Gcn4 AD variants (*Staller et al., 2018*).

Here, we combine quantitative, high-throughput measurements of in vivo activation and in vitro interaction with computational modeling to characterize ADs. We identify 150 ADs in budding yeast and describe their shared sequence attributes. We extend the analysis by training and validating a neural network that predicts new ADs across eukaryotes. Guided by predictions, we design and measure activation of thousands of AD mutants to derive a deeper understanding of the principles underlying activation. Correlating activation with measurements of binding of ADs to Mediator in vitro identifies the key protein interactions driving activation, for which we predict atomic structures using a peptide-docking algorithm. We derive structural features common to AD-Mediator interactions and use them to explain measurements of Mediator recruitment kinetics in vitro. In total, we develop an integrated model for function, sequence determinants, molecular interactions, and kinetic features of TF ADs.

## Results

### A quantitative screen identifies 150 activation domains from all yeast transcription factors

We identified ADs across all TFs in budding yeast with the use of a quantitative, uniform, and high-throughput activation assay. Variability due to protein expression, activation duration, and secondary genetic effects was minimized by fusing domains of interest to a three-part artificial TF (aTF) that (1) is tracked by an mCherry tag, (2) localizes to the nucleus only upon induction with estrogen, and (3) binds uniquely through its mouse DBD in the promoter of a chromosomally-integrated GFP reporter gene (*Figure 1A*; *McIsaac et al., 2013*; *Staller et al., 2018*). In this assay, three known ADs stimulated GFP expression greater than 100-fold upon estrogen treatment (*Figure 1—figure supplement 1A*).

We used a pooled screen to measure activation by 7460 protein segments, each 53 amino acids (aa) in length, that tiled all 164 TFs (identified by Gene Ontology terms) with a step size of 12–13 aa (3.8-fold average coverage; *Figure 1B*). Outgrowth of transformed yeast for 5 days ensured that each cell contained a unique aTF expression plasmid due to mechanisms maintaining it at single-copy levels (Materials and methods) (*Scanlon et al., 2009*). After induction with estrogen, cells with a defined level of aTF expression were selected by mCherry signal and sorted into eight bins based on GFP expression (*Figure 1—figure supplement 1B*). By next-generation sequencing, the distribution of each protein tile across the bins was determined and the mean value was used to calculate activation—namely, the fold increase in GFP relative to background (*Figure 1—source data 1*, Materials and methods). Activation was highly concordant between distinct DNA sequences encoding the same protein fragment (*Figure 1—figure supplement 1C*). GFP distributions of three known ADs in the pooled screen exactly reproduced measurements from individual pure populations (*Figure 1—figure supplement 1D*). Controlling by a defined aTF expression level was critical for precisely measuring activation (*Figure 1—figure supplement 1E–F*). Activation measurements of fragments shared between two independently cloned and assayed libraries were reproducible (Pearson's r = 0.953, *Figure 1—figure supplement 1G*). Over 2,850,000 cells were assayed in total, giving excellent coverage of library elements (approximately 99% of TF tiles were observed in at least 10 cells each).

Our assay exhibited high signal-to-noise: tiles in previously known and newly identified ADs activated nearly 200-fold while 88% of fragments activated less than twofold (*Figure 1C*). Across the library, 451 tiles showed significant activation (p<0.0001 by Z-test). When plotted by protein position, activating tiles clustered into discrete, well-defined ADs (*Figure 1D*). Using a positional activation score, we identified 150 ADs in 96 TFs (*Figure 1E* and *Figure 1—figure supplement 1H*, *Figure 1—source data 2*, Materials and methods). These ADs overlapped 75% of all previously-reported ADs in TFs (*Figure 1—source data 2*). Furthermore, the 53-aa tile length was not limiting, since our screen successfully identified ADs in over 85% of TFs that activated in a previous one-hybrid screen testing full-length proteins (*Figure 1—source data 2*; *Titz et al., 2006*). These results show that our screen is both sensitive and comprehensive.

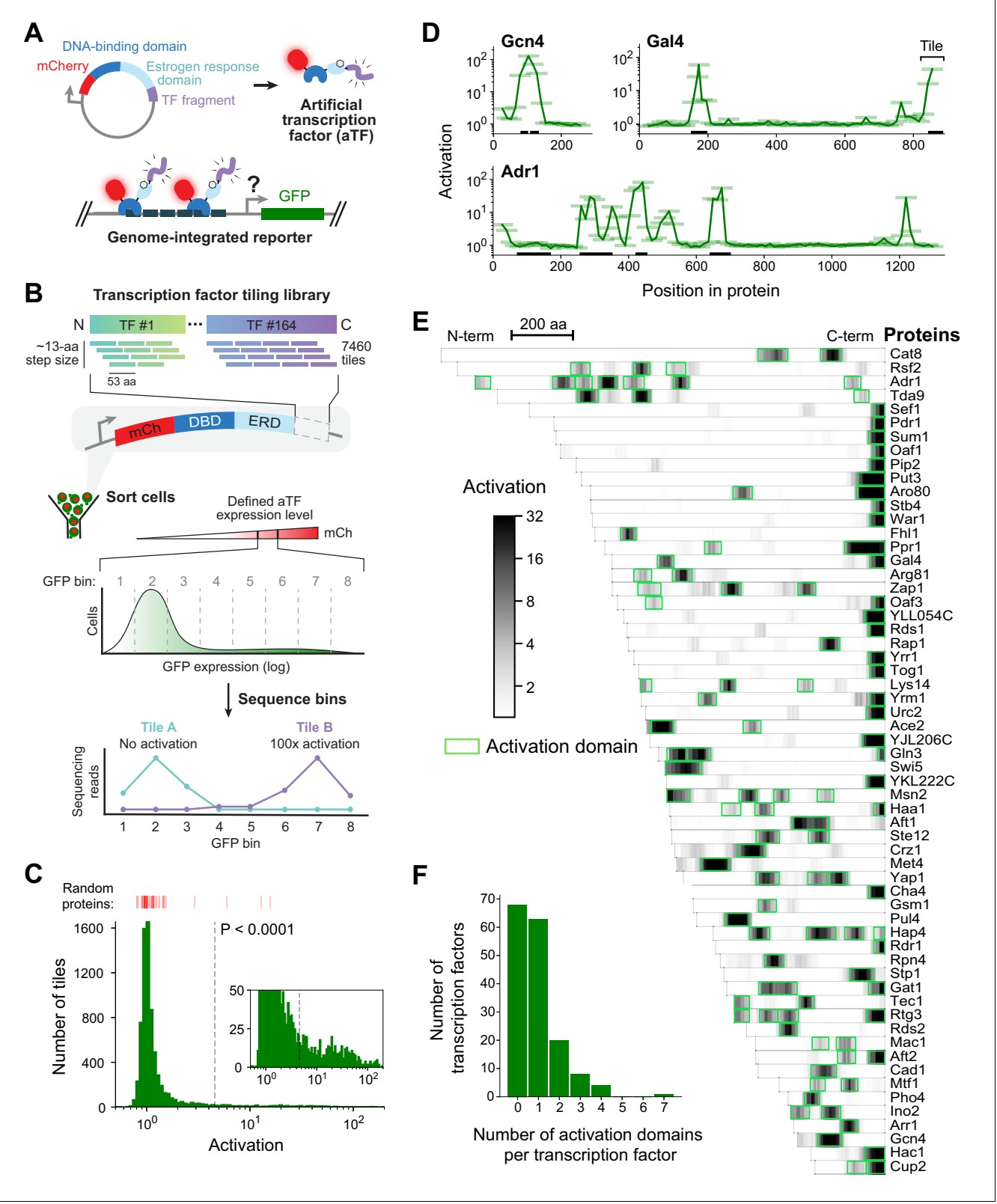

**Figure 1.** A quantitative screen identifies 150 activation domains from all yeast transcription factors. (**A**) Schematic of the activation assay. To measure in vivo activation, we expressed fragments of TF proteins fused to a DNA-binding domain that binds uniquely in the promoter of a genome-integrated GFP reporter gene. This artificial TF (aTF) is tracked by its mCherry tag and localizes to the nucleus only after induction with estrogen. (**B**) Pooled screen for activation domains (ADs). All 164 yeast TFs were tiled by a DNA oligonucleotide library expressing 7460 protein segments, each 53 amino acids (aa)

*Figure 1 continued on next page*

*Figure 1 continued*

in length, with a step size of 12–13 aa (3.8-fold coverage). The library was cloned into the aTF expression plasmid and transformed into yeast cells. Using fluorescence-activated cell sorting, cells with a defined level of aTF expression were selected by mCherry signal and sorted into eight bins based on GFP expression. By next-generation sequencing, the distribution of each protein tile across the bins was determined and the mean value was used to calculate activation—namely, the fold increase in GFP relative to background. See also *Figure 1—figure supplement 1A–D* and *Figure 1—source data 1*. (C) Histogram of activation measured for 7460 tiles spanning all yeast TFs. Dashed line shows cutoff for p-values less than 0.0001 (Z-test). Red bars above the histogram mark activation of 50 random protein sequences. Inset: same histogram, zoomed in. See also *Figure 1—figure supplement 1G*. (D) Activation data for tiles spanning three example proteins. Activating tiles cluster into well-defined ADs when plotted by their protein position. Tiles, 53 aa long, are shown as horizontal bars with a line traced through their centers for clarity. Previously known ADs are marked as black bars along the x-axis. See also *Figure 1—figure supplement 1H*. (E) Heatmap showing the mean activation at each position of the 60 TFs with the strongest ADs. Proteins run left to right from N-terminus to C-terminus and are sorted by length. A scale bar shows 200 aa. ADs annotated in our screen are boxed in green and listed in *Figure 1—source data 2*. The method for annotating ADs is depicted in *Figure 1—figure supplement 1H*. (F) Histogram of the number of ADs in each TF. Of 164 TFs, 68 had no ADs, 63 had a single AD, and 33 had multiple distinct ADs, including up to seven distinct ADs in Adr1 (panel D). See also *Figure 1—figure supplement 1I*.

The online version of this article includes the following source data and figure supplement(s) for figure 1:

**Source data 1.** Data from activation assays.
**Source data 2.** ADs identified in the activation screen and their overlap with previously-known ADs.
**Figure supplement 1.** Methodology, validation, and summary statistics for pooled activation screens.

Three-quarters (112) of ADs identified here were previously unknown (*Figure 1—source data 2*). While 63 TFs contained just a single AD, 33 TFs had multiple, including up to seven distinct ADs in Adr1 (*Figure 1D and F*). C-terminal ADs were common—found in nearly half of all AD-containing TFs—and were stronger and shorter on average than other ADs (*Figure 1—figure supplement 1I*). Consistent with AD function in their native context, TFs that contained ADs upregulated a higher proportion of downstream genes than TFs without ADs (*Figure 1—figure supplement 1J*; *Hackett et al., 2020*). Consistent with the diverse functions of TFs beyond transcriptional activation (e.g. repression), 68 TFs did not contain any ADs (*Figure 1F*).

## Activation strength is primarily determined not by motifs but by acidic and hydrophobic content

ADs were enriched in negative charge and hydrophobic content, and activation was strongest for fragments that were high in both (*Figure 2A* and *Figure 2—figure supplement 1A–B*; Wimley-White hydrophobicity scale) (*Wimley and White, 1996*). Past experiments have clearly shown the importance of bulky hydrophobic residues but presented conflicting evidence for acidic residues in activation (*Brzovic et al., 2011*; *Cress and Triezenberg, 1991*; *Drysdale et al., 1995*; *Jackson et al., 1996*; *Pacheco et al., 2018*; *Staller et al., 2018*; *Sullivan et al., 1998*; *Warfield et al., 2014*). To determine the role of negative charge, we took the strongest-activating fragment from each AD and measured its activation with all acidic residues mutated. Except for a single example that activated just as strongly at +7 net charge without its acidic residues (the Gln3 C-terminal AD), activation was abolished in all ADs, showing definitively that acidic residues are necessary in budding yeast ADs (*Figure 2B* and *Figure 2—figure supplement 1C*). We also noticed that activation of wild-type tiles was more strongly correlated with the number of Asp residues than the number of Glu residues, and hypothesized that Asp promotes activation more strongly than Glu (*Figure 2—figure supplement 1D*). Indeed, ADs with all Glu residues mutated to Asp consistently activated as well as or better than ADs with all Asp residues mutated to Glu (*Figure 2—figure supplement 1E*).

Having identified the important residue types, we looked for sequence motifs that could predict activation. However, AD sequences appeared to share no common motifs upon inspection. The 9aaTAD motif, originally proposed based on yeast ADs (*Piskacek et al., 2007*), predicted ADs poorly: only 10% of 9aaTAD-containing tiles activated in our assay, which was not significantly more predictive than scrambled versions of the motif (*Figure 2C*). A de novo search for motifs using DREME also found none that were shared by more than two ADs (*Bailey, 2011*). Evidently, ADs are diverse in sequence and not predictable from simple motifs.

The lack of motifs suggests that activation is determined not by positions of individual residues but by distributed or redundant features. We pursued this point by measuring activation of a panel

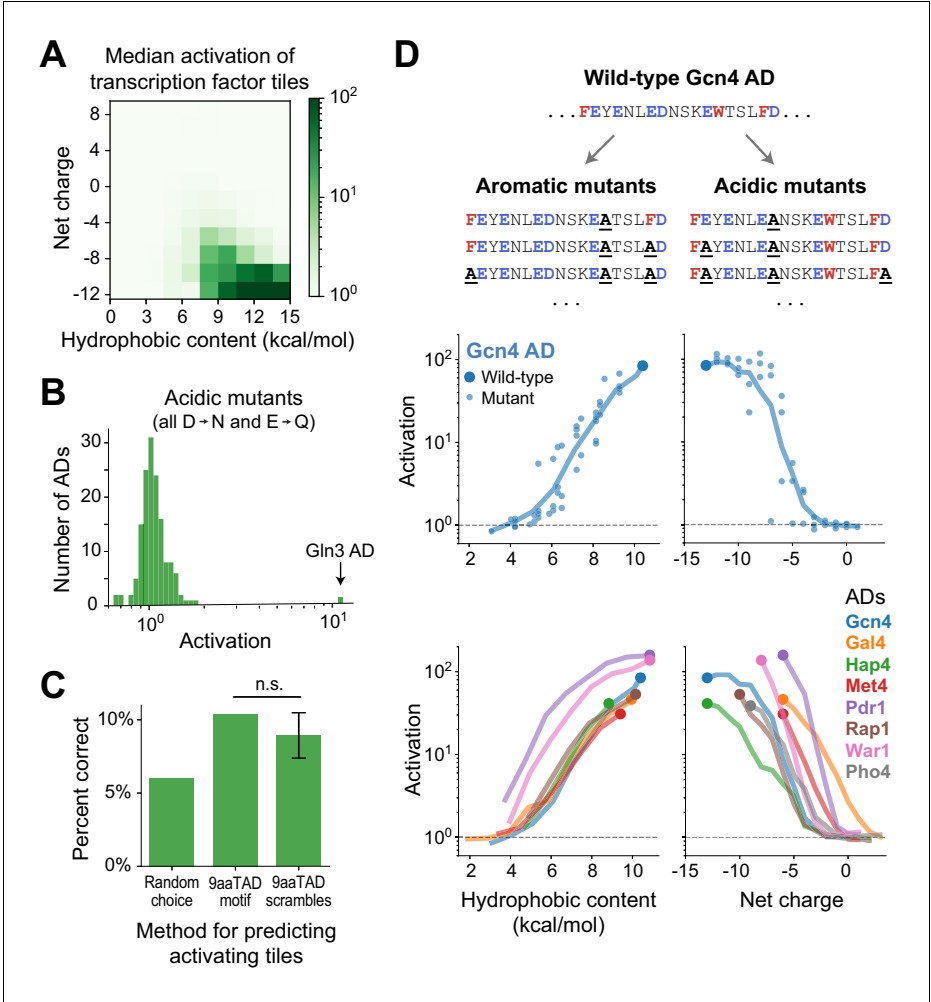

**Figure 2.** Activation strength is primarily determined not by motifs but by acidic and hydrophobic content. (**A**) Tiles were binned by their hydrophobic content and net charge and the median activation of each bin is displayed in color. Activation was strongest for tiles high in both acidic and hydrophobic content. Hydrophobicity was computed using the Wimley-White scale (***Figure 2—figure supplement 1B***) and charge was computed by counting Asp, Glu, Arg, and Lys residues. (**B**) From each AD, activation of the strongest-activating fragment with all acidic residues mutated (Asp to Asn and Glu to Gln). Except for the Gln3 C-terminal AD, which activated just as strongly at +7 net charge without its acidic residues, activation was abolished in all ADs. See also ***Figure 2—figure supplement 1C–E***. (**C**) Just 10% of tiles containing the 9aaTAD motif activated, which was only 1.7-fold better than guessing activating tiles at random and not significantly more predictive than scrambled versions of the motif. (n.s., not significant.) (**D**) We varied the acidic and hydrophobic content of eight ADs by mutating successively larger subsets of aromatic (left) or acidic (right) residues. (*Top*) Example mutant sequences for a segment of the Gcn4 AD, with activation of the wild-type (large dot) and mutants (small dots) plotted below as a function of hydrophobic content or net charge. Lines trace a moving average of activation. (*Bottom*) Average activation as a function of hydrophobic content or net charge, for all eight ADs tested. Activation of all individual mutants is shown in ***Figure 2—figure supplement 1H***. The Pho4 AD contains only two aromatic residues so its aromatic mutants are not shown.

The online version of this article includes the following figure supplement(s) for figure 2:

**Figure supplement 1.** Mutagenesis of hydrophobic and acidic residues in known and newly-discovered ADs.

of designed mutants of eight ADs. Indeed, 94% of scanning single-amino acid mutants affected activation by less than twofold, and all still activated more than 10-fold (***Figure 2—figure supplement 1F–G***). To produce larger effects, we varied the acidic and hydrophobic content of the ADs by mutating successively larger subsets of acidic or aromatic residues (***Figure 2D***, Materials and

methods). Activation was gradually, and finally completely, eliminated as either characteristic was reduced. It is noteworthy that the Gcn4 AD, a focus of many past studies, uniquely tolerated mutation of up to six acidic residues. Most notably, activation by acidic and hydrophobic mutants were related to the number, rather than position, of mutated residues (*Figure 2D* and *Figure 2—figure supplement 1H*). Thus, activation strength is determined in large part by acidic and hydrophobic content, not by motifs.

## A deep learning model, termed PADDLE, predicts the location and strength of acidic activation domains in yeast and human

Since motifs predicted ADs poorly, we turned to more sophisticated approaches using machine learning. We first trained a simple neural network to predict activation based solely on tiles' amino acid (aa) composition (Materials and methods). When tested on TF tiles withheld from training, this neural network performed remarkably well, explaining 66% of observed variation (*Figure 3—figure supplement 1A*). However, this algorithm was limited by its lack of information about residue positions. To experimentally disentangle the contributions of aa positioning and aa composition, we measured activation of eight scrambled sequences from each of the eight ADs tested before. Despite their shared abundance of acidic and hydrophobic residues, these mutants spanned a wide range of activation strengths, and some mutants activated stronger than wild-type while others failed to activate entirely (*Figure 3A*). Clearly, full sequence information beyond aa composition is needed to predict activation.

We therefore trained a deep convolutional neural network (CNN) to predict activation based on protein sequence, predicted secondary structure, and predicted disorder (Materials and methods). CNNs evaluate sequences by hierarchically integrating matches to a suite of learned patterns and have recently found great success in many genomic prediction tasks (*Ching et al., 2018*; *Kelley et al., 2016*; *Wang et al., 2016*). Our Predictor of Activation Domains using Deep Learning in Eukaryotes, or 'PADDLE', explained 81% of observed variation in TF tiles withheld from training (alternatively, area under the precision-recall curve of 0.805; *Figure 3—figure supplement 1B–C*, *Figure 3—source data 1*), markedly better than the aa composition-based predictor. De novo, PADDLE accurately predicted the activation strength of (1) new ADs within TFs omitted from training ($R^2$ score = 0.81 across tiles from 22 TFs; example predictions for Arg81 in *Figure 3B*); (2) scrambled AD sequences, despite their identical amino acid composition ($R^2$ score = 0.61); and (3) 232 mutants and 178 orthologs of the Pdr1 AD ($R^2$ score = 0.90) (*Figure 3—figure supplement 1B*). Altogether, the performance of PADDLE was validated across a wide range of both wild-type and mutant sequences in yeast.

Classic experiments showed that many acidic ADs retained activity even when transferred between yeast and animals (*Fischer et al., 1988*; *Struhl, 1988*), so we investigated whether PADDLE could identify ADs in human TFs that would activate in human cells. Using PADDLE, we predicted 236 high-strength and 366 moderate-strength ADs, together spanning 462 (27%) human TFs (*Figure 3C*, *Figure 3—source data 1*). These predicted ADs overlapped many known ADs of TFs from diverse families, including p53, NFkB, Myc, Klf4, Fos, PPARA, SREBF1, E2F proteins, and the glucocorticoid receptor (*Figure 3—figure supplement 1D*). PADDLE also predicted 41 high-strength and 45 moderate-strength ADs from among 419 transcription-regulating viral proteins (*Figure 3—source data 1*; *Liu et al., 2020*). We randomly selected 25 high-strength predicted ADs from human TFs and measured their activation individually using a luciferase reporter in HEK293T cells. Remarkably, 23 domains (92%) activated luciferase expression (*Figure 3D*). While other classes of human ADs, such as glutamine-rich or proline-rich ADs, are not predictable by PADDLE since they are not present in *S. cerevisiae*, acidic activation mechanisms are evidently conserved and the patterns learned by PADDLE are generalizable across eukaryotes.

## PADDLE identifies the core regions and key residues of every activation domain

To uncover the principles of acidic activation learned by PADDLE, we analyzed predictions on designed mutant sequences and developed hypotheses, which we then experimentally tested in a second library. We noticed from predictions on sequences less than 30 aa that yeast ADs contained short, independently activating regions, which we term core ADs (cADs) (*Figure 3—figure*

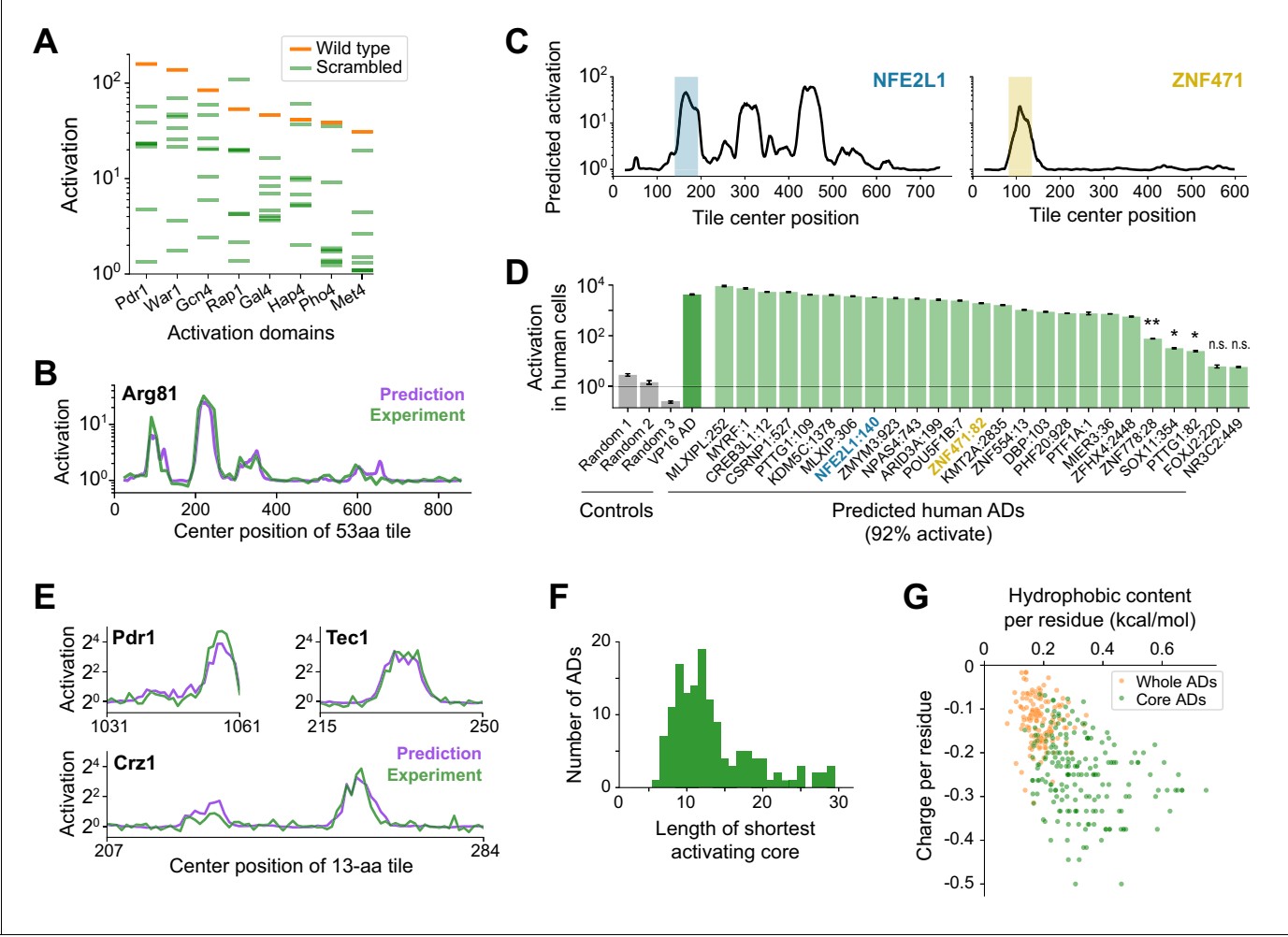

**Figure 3.** A deep learning model, termed PADDLE, predicts the location and strength of acidic activation domains in yeast and human. (A) Activation of wild-type (orange) and eight scrambled sequences (green) for each of eight ADs. (B) De novo PADDLE predictions (purple) and experimentally measured activation (green) for Arg81 are plotted by protein position. Predictions were run on 53-aa tiles in 1-aa steps and smoothed with a 9-aa moving average. See also *Figure 3—figure supplement 1A–C* and *Figure 3—source data 1*. (C) Example PADDLE predictions on 53-aa tiles spanning two human TFs. One predicted AD from each TF, marked by the colored shading, was tested experimentally. See also *Figure 3—figure supplement 1D* and *Figure 3—source data 1*. (D) PADDLE accurately predicts human ADs. We randomly selected 25 high-strength predicted ADs from human TFs and measured their activation individually using a luciferase reporter in HEK293T cells. Relative to three random sequence controls, 23 domains (92%) activated luciferase expression. The VP16 AD was used as a positive control. Error bars show standard deviation of technical triplicates. (*p<0.01; **p<0.0005; n.s., not significant.). (E) PADDLE predictions (purple) and experimentally measured activation (green) of 13-aa tiles in 1-aa steps spanning three ADs. Both predictions and experiments identified at least one significantly activating 13-aa tile within each of 10 ADs (*Figure 3—figure supplement 1E–F*). (F) PADDLE was used to identify, within each AD, the shortest core region predicted to activate on its own. A histogram of their lengths is plotted. The minimal region for activation can be localized to within 20-aa core in 85% of ADs. (G) The hydrophobic content and charge per residue of whole ADs (orange) and predicted core ADs (green).

The online version of this article includes the following source data and figure supplement(s) for figure 3:

**Source data 1.** ADs predicted in human and virus proteins and core ADs predicted in yeast.

**Figure supplement 1.** Validation of neural network models and PADDLE predictions on human TFs and yeast core ADs.

*supplement 1E*). As an experimental test, we measured in vivo activation of 13-aa-long fragments tiling ten ADs in 1-aa steps. Predictions proved highly accurate ($R^2$ score = 0.84, *Figure 3—figure supplement 1F*), and both predictions and experiments identified at least one significantly activating 13-aa cAD within each AD (*Figure 3E*; p<0.001).

Across all yeast TFs, PADDLE predicted that the minimal region for activation could be localized to within a 20-aa cAD in 85% of ADs (*Figure 3F*, *Figure 3—source data 1*). These cADs still shared

no motifs (*Figure 3—figure supplement 1G*), but instead had an especially high density of acidic and hydrophobic residues, even more so than the full ADs (*Figure 3G*). To quantify the contribution of each residue to activation, we predicted the impact of mutating each position in each cAD to alanine (*Figure 3—figure supplement 1H*). Single-aa mutants in these short cADs had a large predicted impact on activation, unlike in the longer 53-aa ADs (*Figure 2—figure supplement 1F–G*), and showed clear trends when grouped by residue identity. Mutating the bulkiest hydrophobic residues led to the greatest decreases in predicted activation, and the three most important were Trp, Phe, and Leu (*Figure 3—figure supplement 1I*). Mutating acidic or basic residues had opposite but comparable effects on predicted activation. Together, these analyses identify the regions and residues across ADs that contribute most to activation.

## A high density of acidic and hydrophobic residues is sufficient for most core ADs to activate, but some require a defined sequence and alpha helical structure

Having identified the core AD regions, we sought to understand the sequence properties that drive their activation. We focused on the 28 strongest 13-aa cADs (*Figure 3—source data 1*), and first experimentally quantified the importance of their aa composition by comparing each cAD with 33 random scrambles of its sequence. Remarkably, only nine cADs (32%) showed activation by the wild-type sequence greater than threefold greater than the average activation by scrambled sequences (*Figure 4A*). In these nine cADs, maximal activation evidently requires a specific positioning of residues. On the other hand, 18 cADs (64%) showed activation by the wild-type sequence within twofold of the scrambled sequence average, indicating that their activation is primarily determined by aa composition and not by a unique positioning of residues. The wild-type-to-scramble ratio was not correlated with wild-type activation strength (*Figure 4—figure supplement 1A*, Pearson's r = 0.32, p=0.09). Instead, cADs with the greatest hydrophobic content had the lowest wild-type-to-scramble ratios (*Figure 4B*, Pearson's r = −0.51, p=0.006). Thus, the majority of cADs studied here activate by a composition-driven mechanism that exploits an excess of bulky hydrophobic residues.

Alpha-helical folding is thought to be important for activation, which is at odds with the prevalence of composition-driven cADs. We directly determined whether the 28 strongest 13-aa cADs require helical folding by measuring the impact of inserting a helix-breaking proline. Within the seven central positions of each 13-aa cAD, we individually mutated each non-hydrophobic residue to alanine or proline and asked whether proline inhibited activation relative to alanine (*Figure 4C*). Proline mutations inhibited activation more than threefold over alanine in nine ADs (32%), including up to an 11-fold drop in activation in the Tda9 cAD. However, proline mutations inhibited activation by less than twofold in 16 cADs (58%). These effects were consistent between different positions within the same cAD (*Figure 4—figure supplement 1B*). Interestingly, all three cADs that contained a basic residue (Pul4, Tda9, and Rsf2:586) were strongly inhibited by proline, suggesting that a helix is necessary to position their inhibitory positive charge away from the coactivator binding interface. Most notably, the magnitude of proline disruption was tightly correlated with the wild-type-to-scramble ratio (*Figure 4D*, Pearson's r = 0.842), showing that constraints on structure and sequence, or a lack thereof, were closely linked. Thus, composition-driven cADs typically do not need to form a helix and likely sample disordered conformations, while cADs that require their wild-type sequence typically also require helical folding, employing a structure-driven mechanism.

To understand these two contrasting mechanisms, we studied them in a simplified system of artificial cADs that used only the four most activation-promoting residues. We examined 9-aa sequences (denoted '9mers') consisting entirely of only two types of amino acids each—Asp and either Leu, Phe, or Trp—so that all possible such sequences ($3 \times 2^9 = 1536$) could be systematically assayed (*Figure 4E*). As expected, activation required a balance of hydrophobic and acidic residues (*Figure 4—figure supplement 1C*), so we focused analysis on aa compositions with the strongest median activation: 9mers with five or six hydrophobic residues. Within these, nearly all Phe 9mers and Trp 9mers activated regardless of their sequence (*Figure 4F*), characteristic of a composition-driven mechanism and consistent with the highly hydrophobic nature of Phe and Trp residues. However, Leu 9mers spanned a wide range of activity with only a minority that were strongly activating, characteristic of structure-driven activation requiring a specific ordering of residues.

To see what allows only certain Leu 9mers to activate, we examined all possible short motifs containing two or three Leu residues. The only motif strongly associated with activation was LxxLL

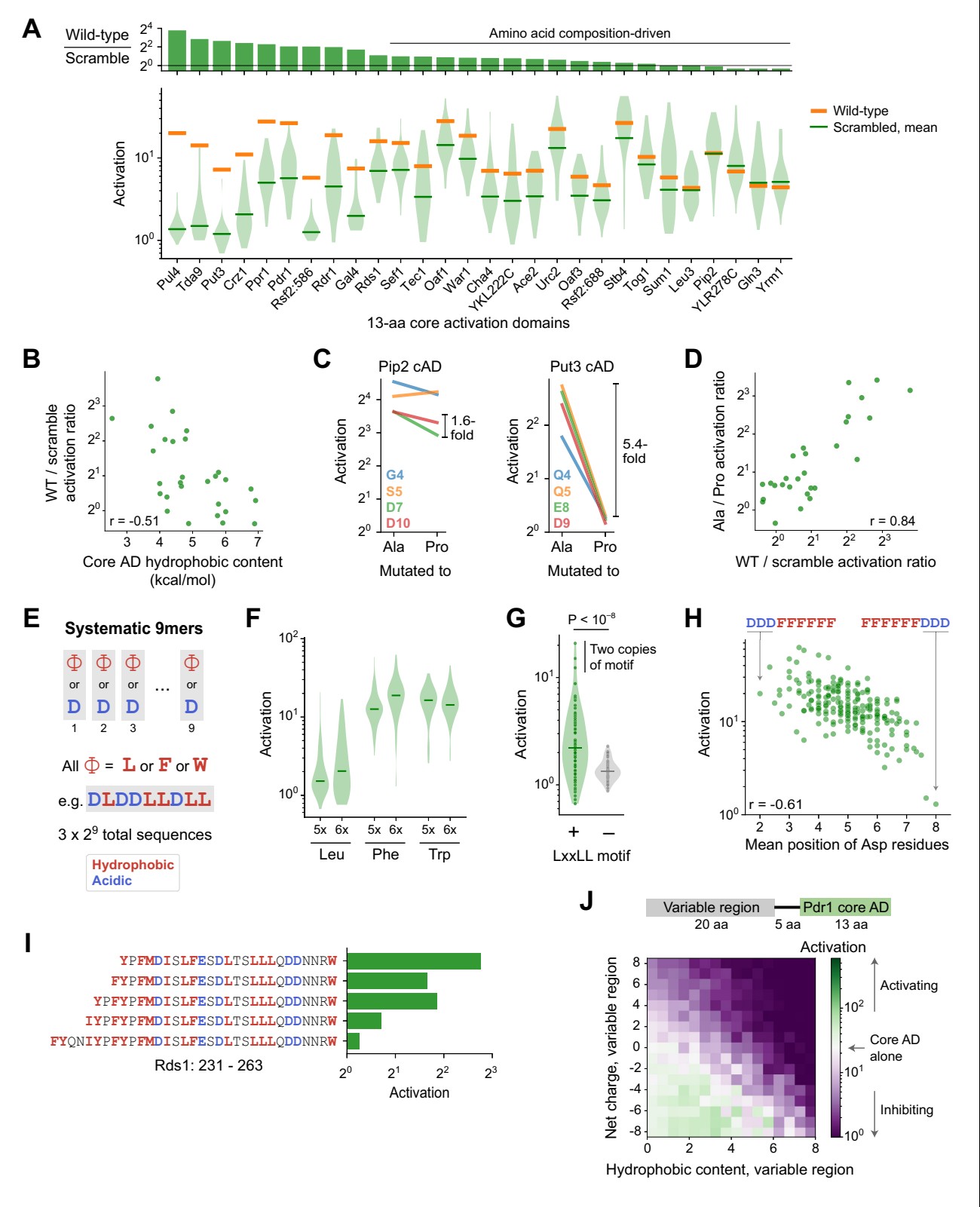

**Figure 4.** Sequence and structural determinants of activation domains. (**A**) To quantify the importance of aa composition, we measured activation of 33 scrambled sequences for each of the 28 strongest 13-aa core ADs (cADs). (*Top*) Activation of each wild-type cAD divided by the mean activation of its scrambled mutants. Eighteen cADs (64%) showed activation by the wild-type sequence within twofold of the scrambled sequence average, indicating that their activation is primarily determined by aa composition and not by a unique positioning of residues. (*Bottom*) Activation of each wild-type cAD

*Figure 4 continued*

(orange bars) and the distribution of activation by scrambled mutants (violin plot), with the mean activation of mutants shown by the green bars. See also *Figure 4—figure supplement 1A*. (B) cADs with the greatest hydrophobic content had the lowest wild-type-to-scramble ratios (Pearson's r = −0.51, p=0.006). (C) To directly determine whether the 28 strongest 13-aa cADs require helical folding, we individually mutated each non-hydrophobic residue to Ala or Pro and asked whether Pro inhibited activation relative to Ala. Activation of the Pip2 cAD (left) was not disrupted by Pro mutation and showed at most a 1.6-fold drop compared to Ala mutations. In contrast, activation of the Put3 cAD (right) was abolished by all Pro mutations, up to a 5.4-fold drop compared to Ala mutations. Effects of mutations for all 28 cADs are shown in *Figure 4—figure supplement 1B*. (D) Across cADs, the wild-type-to-scramble ratio (horizontal axis) was correlated with the maximal fold-drop in activation resulting from Pro mutation (vertical axis). Pearson's r = 0.84. (E) A simplified system of artificial cADs that uses only the four most activation-promoting residues. Namely, 9-aa sequences ('9mers') consisting entirely of only two types of amino acids each: Asp and either Leu, Phe, or Trp. This way, all possible sequences ($3 \times 2^9 = 1536$) could be systematically assayed. (F) Distributions of activation by 9mers consisting of five or six Leu, Phe, or Trp residues, grouped by aa composition, are shown as violin plots. Activation by all 9mers, grouped by aa composition, is shown in *Figure 4—figure supplement 1C*. (G) The LxxLL motif is significantly predictive of activation in 9mers with five Leu residues ($p<10^{-8}$ by Kolmogorov-Smirnov test). The three strongest-activating sequences each contain two copies of the motif. This was the only motif strongly associated with activation of these Leu 9mers; effects for all motifs tested are shown in *Figure 4—figure supplement 1D*. Similar motifs were also significantly but more weakly associated with Phe 9mers and Trp 9mers (*Figure 4—figure supplement 1E*). (H) Activation by 9mers with five or six Phe residues was correlated with the average position of their acidic residues (Pearson's r = −0.61). Most dramatically, DDDFFFFFF activated 20-fold while its reverse sequence FFFFFFDDD activated just 1.3-fold. (I) Residues 239–263 alone in Rds1 activated 6.8-fold, but extending the sequence by eight aa, five of them hydrophobic (red), abolished activation. (J) (*Top*) To systematically quantify inhibition, we measured the effect on activation when the Pdr1 cAD was placed next to a library of 2177 random 20-aa sequences chosen to span a wide range of net charge and hydrophobic content. (*Bottom*) Sequences were binned by the net charge and hydrophobic content of the variable region and mean activation is plotted in color. Activation comparable to the Pdr1 cAD alone is shown in white, with stronger and weaker activation shown in green and purple, respectively.

The online version of this article includes the following figure supplement(s) for figure 4:

**Figure supplement 1.** Additional activation data for and analysis of core AD mutants and systematic 9mers.

---

(*Figure 4G* and *Figure 4—figure supplement 1D*; $p<10^{-8}$ by Kolmogorov-Smirnov test). This motif was strictly required for activation in these 9mers, present in two copies in the three strongest 9mers, found in two wild-type structure-driven cADs (Put3 and Crz1), and previously described in ADs of many nuclear receptors and other TFs (*Plevin et al., 2005*). This motif also suggests an explanation for why helical folding was necessary for cADs with less hydrophobicity: by placing Leu residues along one face of an alpha helix, the motif efficiently forms a hydrophobic binding surface.

The analogous FxxFF and WxxWW motifs were also significantly but more weakly associated with activation of Phe 9mers and Trp 9mers (*Figure 4—figure supplement 1E*). However, their strongest predictor of activation was the average position of acidic residues, with N-terminal skew highly favored (*Figure 4H*). Most dramatically, DDDFFFFFF activated 20-fold while its reverse sequence FFFFFFDDD activated just 1.3-fold. One role of N-terminal negative charge could be to neutralize the macroscopic dipole that arises from alpha helix backbone hydrogen bonds, stabilizing helical conformations (*Rocklin et al., 2017*). Even though a helix is not necessary for composition-driven activation, this effect could reduce the entropic cost of binding to coactivators by promoting transient folding.

## Activation domains are strongly inhibited by nearby clusters of hydrophobic residues

Despite the importance of hydrophobic residues for activation, PADDLE also predicted that some TFs contained cADs whose activation was inhibited by nearby clusters of hydrophobic residues. We confirmed this experimentally for Rds1, in which residues 239–263 alone activated 6.8-fold but extending the sequence by eight aa, five of them hydrophobic, abolished activation (*Figure 4I*). Inhibition by hydrophobic residues also explains the weak activation of some scrambled 53-aa ADs (*Figure 3A*, *Figure 4—figure supplement 1F*).

To systematically quantify inhibition, we measured the effect on activation when the Pdr1 cAD was placed next to a library of 2177 random 20-aa sequences chosen to span a wide range of net charge and hydrophobic content (*Figure 4J*). As expected, negative charge with moderate hydrophobicity bolstered activation. Positive charge without hydrophobicity inhibited activation, but even variable regions with +8 charge (+4 net charge) still activated an average of 4.4-fold. Therefore, the local clustering of negative charges near hydrophobic residues renders the Pdr1 cAD resistant to global changes in net charge, suggesting that interactions between opposite charges within the

peptide do not fully prevent binding to coactivators. In contrast, hydrophobic residues at high density inhibited activation completely. This suggests that inhibitory hydrophobic clusters interact with hydrophobic residues in the cAD, completely preventing their interaction with coactivators. Consistent with two independent mechanisms of inhibition, basic and hydrophobic residues together reduced activation additively. In total, these experiments mapped the biochemical and structural properties that drive both activation and inhibition.

## The large majority of activation domains bind Mediator, and its recruitment is a key driver for activation

The surprising ability of simple biochemical features to explain a large and diverse set of ADs suggests that these features may also drive interactions with coactivator complexes. To gain a mechanistic understanding of the AD-coactivator contacts that underlie activation, we mapped the interactions of wild-type and mutant ADs with Mediator. We focused on Med15, the primary subunit of Mediator targeted by many ADs in budding yeast, which contains four tandem activator-binding domains (ABDs) (*Herbig et al., 2010*; *Jedidi et al., 2010*; *Thakur et al., 2008*). We isolated the N-terminal portion of Med15 consisting of its four ABDs (hereafter just called 'Med15'), confirmed its in vitro interaction with Gcn4, and used it for binding experiments (*Figure 5—figure supplement 1A*).

To measure binding in high-throughput, we used mRNA display (*Takahashi and Roberts, 2009*), expressing our library of TF tiles as a pool of protein fragments covalently tagged with their mRNA sequences, and using this pool in Med15 pull-down experiments (*Figure 5A*). Direct counts of Med15-bound and input protein molecules were obtained by amplifying and sequencing their mRNA tags and using unique molecular identifiers (UMIs) to remove PCR duplicates. Finally, the fractional pull-down of each protein fragment was computed by its abundance in the Med15-bound sample relative to input, normalized to total library concentrations measured by qPCR (Materials and methods). Pull-down measurements were reproducible between replicates (Pearson's r > 0.90) and consistent between identical protein fragments encoded by distinct but synonymous mRNA sequences (*Figure 5—figure supplement 1B–C*). Fragments known to bind Med15 enriched up to 17-fold above random sequence controls and 1000-fold higher than in mock pull-downs with beads alone (*Figure 5—figure supplement 1D*, *Figure 5—source data 1*).

Binding to Med15 was widespread among ADs and coincident with activation activity. In total, 324 tiles bound Med15 significantly more than negative controls (p<0.001 by Z-test, *Figure 5—figure supplement 1E*, Materials and methods). When plotted by protein position, Med15-binding tiles clustered into 153 discrete domains, the vast majority of which were not previously known to bind Mediator (*Figure 5B*, *Figure 5—source data 1*). ADs and Med15-binding domains overlapped substantially: 73% of ADs bound Med15, and 71% of Med15-binding domains functioned as ADs (*Figure 5C*). Moreover, ADs that did not bind Med15 tended to activate weakly (*Figure 5—figure supplement 1F*), so it is possible that many bind Med15 at levels below the detection limit of our pull-down assay, which was less sensitive than our activation assay (compare *Figure 1C* and *Figure 5—figure supplement 1E*). Just as with activation, high acidic and hydrophobic content were associated with Med15 binding (*Figure 5—figure supplement 1G*). The Gln3 AD that did not require acidic residues also did not bind Med15 substantially (2.1-fold enrichment), suggesting that it activates through other mechanisms.

Most notably, activation strength and Med15 binding of tiles were directly proportional (*Figure 5D*). To test this relationship further, we measured Med15 binding of the set of AD mutants in which aromatic residues were systematically removed (*Figure 2D*). In every case, Med15 binding and activation were strongly correlated (*Figure 5E*), showing that the hydrophobic features that drive activation similarly underlie Med15 binding. Moreover, the ability for binding of a single protein in vitro to quantitatively explain activation in vivo suggests that Mediator recruitment is a key driver for gene activation and determined largely by AD binding to its Med15 subunit.

ADs can bind different coactivators, though the extent of such interaction is unknown. To explore the possibility, we also examined binding of TF tiles to TFIID, a key factor in the transcription of nearly all genes (*Warfield et al., 2017*). Yeast TFIID contains three known AD-binding subunits, Taf4, Taf5, and Taf12, which form a subcomplex with Taf6 and Taf9 (*Layer et al., 2010*; *Wright et al., 2006*). We isolated this TFIID subcomplex and confirmed its interaction with the VP16 AD (*Figure 5—figure supplement 1H*). Pull-downs of the TF tiling library with the TFIID subcomplex

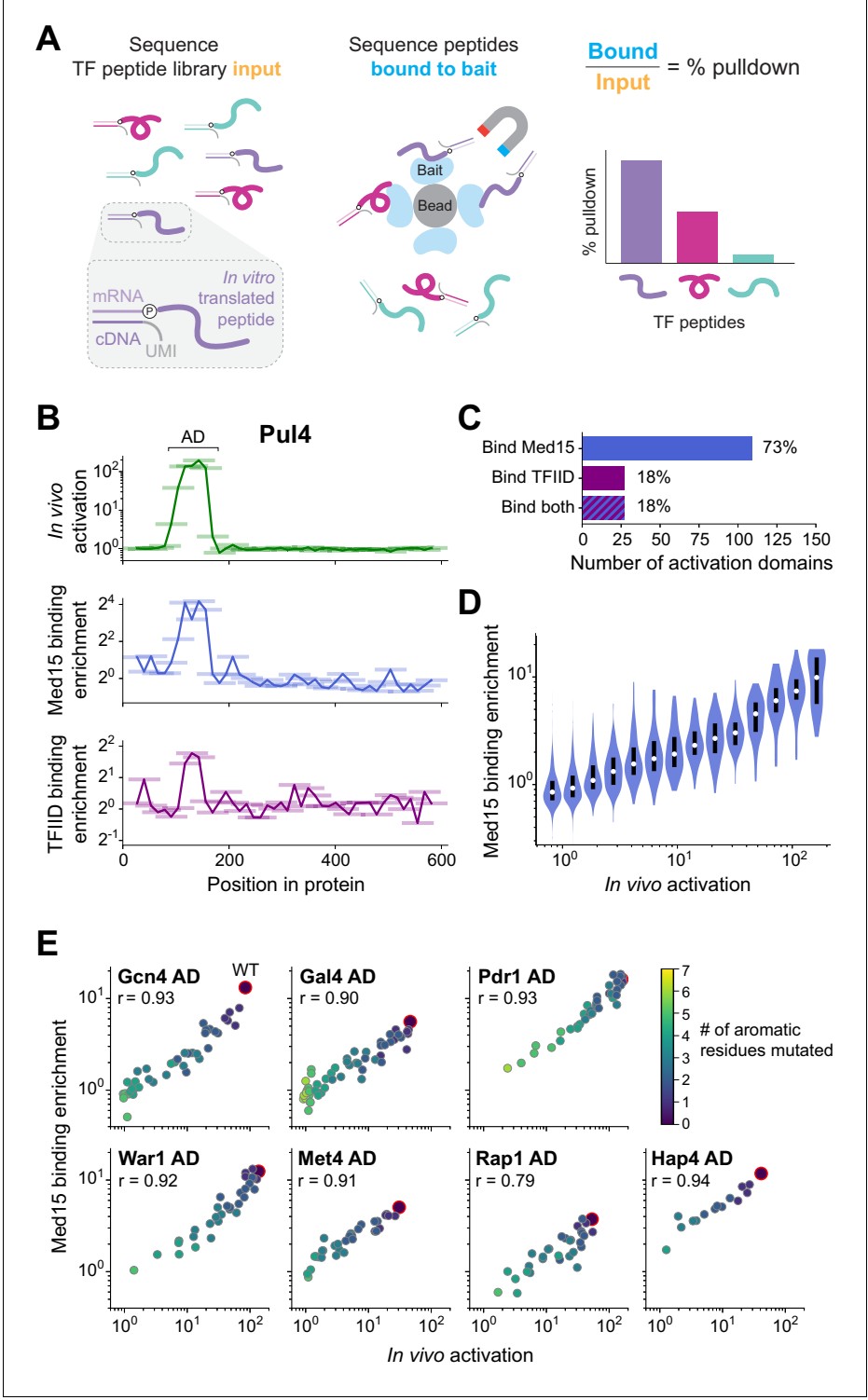

**Figure 5.** The large majority of activation domains bind Mediator, and its recruitment is a key driver for activation. (**A**) To measure binding in high-throughput, we used mRNA display, expressing our library of TF tiles as a pool of protein fragments covalently tagged with their mRNA sequences (left), and using this pool in pull-down experiments (middle). Direct counts of bound and input protein molecules were obtained by amplifying and sequencing their mRNA tags and using unique molecular identifiers (UMIs) to remove PCR duplicates. Finally, the fractional pull-down of each protein fragment was computed by its relative abundance in the bound sample versus input, normalized to total library concentrations measured by qPCR (right). (**B**) For all tiles spanning Pul4, in vivo

*Figure 5 continued on next page*

*Figure 5 continued*

activation (green), in vitro Med15 binding enrichment (blue), and in vitro TFIID subcomplex binding enrichment (purple) are plotted. Tiles, 53 aa long, are shown as horizontal bars with a line traced through their centers for clarity. The Pul4 AD binds both Med15 and TFIID. Binding enrichments of all tiles are in *Figure 5—source data 1*. (C) Number and percentage of ADs that bound Med15, TFIID subcomplex, or both in pull-down experiments. ADs and Med15-binding domains overlapped substantially. All ADs that bound TFIID also bound Med15. See also *Figure 5—figure supplement 1K* and *Figure 5—source data 1*. (D) Violin plot of tiles' Med15 binding enrichment, grouped by their in vivo activation. A white dot labels the median of each bin and a black bar marks the 25th to 75th percentile interval. See also *Figure 5—figure supplement 1F*. (E) We measured Med15 binding of the set of AD mutants in which aromatic residues were systematically removed (*Figure 2D*). Med15 binding and activation are plotted with wild-type sequences outlined in red, the number of mutated residues shown in color, and Pearson's r values displayed.

The online version of this article includes the following source data and figure supplement(s) for figure 5:

**Source data 1.** Data from pull-down assays and list of Med15-binding domains.
**Figure supplement 1.** Methodology, validation, and summary statistics for Med15 and TFIID subcomplex pull-down screens.

---

enriched fragments up to 6.7-fold above random sequence controls (*Figure 5—figure supplement 1I–J*, *Figure 5—source data 1*). In total, 27 ADs (18%) across 26 TFs bound the TFIID subcomplex (p<0.001), greatly expanding its repertoire of known interactions (*Figure 5B–C*). However, all ADs that bound TFIID also bound Med15, and despite the identical pull-down conditions, ADs bound more weakly to TFIID than to Med15 (*Figure 5—figure supplement 1K*). While more TFIID interactions may be discoverable if assay sensitivity can be further optimized, these results suggest that TFIID has lower affinity for most ADs and provide no evidence for its specific targeting as a coactivator. Since TFIID binding by ADs is redundant and less frequent, its role in most TF-directed activation is secondary to that of Mediator.

## Med15 uses a shape-agnostic, fuzzy interface to bind diverse activation domain sequences

What structural features of Med15 enable its promiscuous yet functional interaction with diverse AD sequences? In the best studied example, the Gcn4 AD interacts with hydrophobic patches and basic residues on multiple Med15 activator-binding domains (ABDs) in a large number of binding poses to form a 'fuzzy' complex (*Brzovic et al., 2011*; *Tuttle et al., 2018*). We addressed the question by using FlexPepDock, a peptide docking algorithm from the Rosetta suite (*Leaver-Fay et al., 2011*; *Raveh et al., 2011*), to systematically build structural models of ABD-AD interactions, made computationally tractable by our identification of short cADs. Our modeling focused on two ABDs with known structures: the KIX domain, which has homology to human Med15 and the p300/CBP coactivator family, and ABD1 (*Figure 6A*; *Brzovic et al., 2011*; *Thakur et al., 2008*). To sample diverse sequences, aa composition, and secondary structure, we modeled interactions of the 28 13-aa cADs described above (*Figure 4A–D*) with the KIX domain and ABD1. For each interaction, 50,000 candidate structural models were sampled and ranked by the Rosetta energy score, and the 10 best-scoring models from each interaction were used in subsequent analyses (*Figure 6—source data 1* and *Figure 6—source data 2*).

We validated our structural modeling by comparison to experimental data in two ways. First, KIX domain residues previously shown by NMR to be important for interacting with the Pdr1 AD were recapitulated by the best-scoring model of this interaction (*Figure 6—figure supplement 1A*; *Thakur et al., 2008*). Second, the degree of alpha helix formation across the 28 cADs was consistent with the effect of proline insertion on in vivo activation (*Figure 6—figure supplement 1B*). As expected, cADs that were inhibited by proline predominantly bound both domains as an alpha helix, and conversely cADs that bound both domains in disordered conformations were minimally affected by proline insertion. Some cADs that were minimally affected by proline insertion were modeled as alpha helical, possibly because disordered regions are underrepresented in protein structure databases on which Rosetta was trained.

A unifying feature of the interactions was the prominent role of hydrophobic contacts at the protein interface. Hydrophobicity-driven interactions between proteins often employ high shape

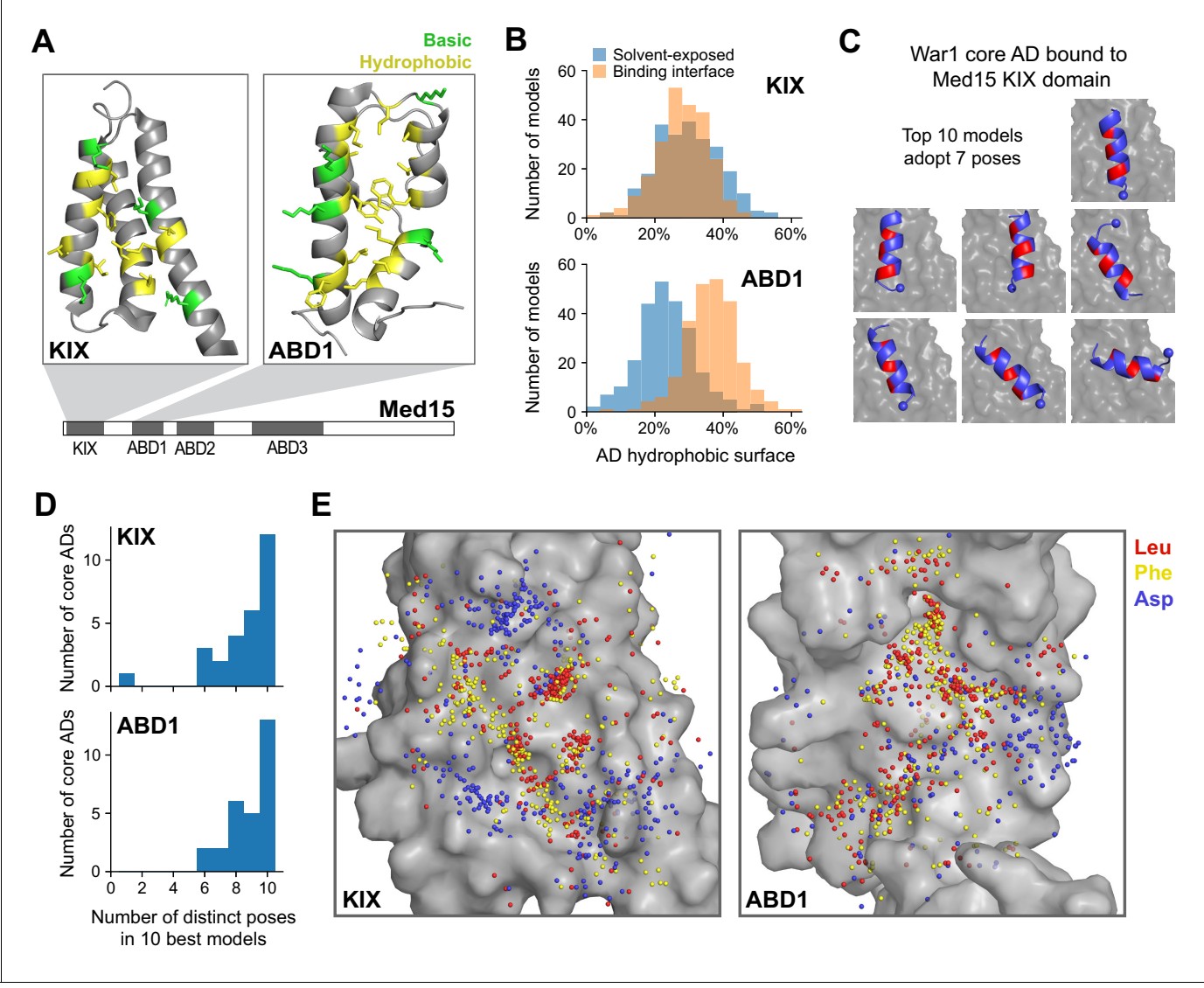

**Figure 6.** Med15 uses a shape-agnostic, fuzzy interface to bind diverse activation domain sequences. (**A**) We used a Rosetta peptide docking algorithm (*Raveh et al., 2011*) to build structural models of the 28 13-aa core ADs (cADs) described above (*Figure 4A–D*) interacting with two activator-binding domains (ABDs) of Med15, the KIX domain and ABD1. Structures of these domains are shown, with the hydrophobic (yellow) and basic (green) residues that form the AD-binding surfaces displayed (*Thakur et al., 2008*; *Herbig et al., 2010*). For each interaction, 50,000 candidate structural models were sampled and ranked by the Rosetta energy score, and the 10 best-scoring models from each interaction were used in subsequent analyses (*Figure 6— source data 1* and *Figure 6—source data 2*). See also *Figure 6—figure supplement 1A*. (**B**) Histograms summarizing the structural features of the 10 best-scoring models of all cADs bound to the KIX domain and ABD1. The horizontal axis denotes the percentages of the total hydrophobic surface of cAD residues that is solvent-exposed (blue) or at the binding interface (orange). See also *Figure 6—figure supplement 1D*. (**C**) The 10 best-scoring models of the War1 cAD (blue) bound the KIX domain (gray surface) with its helix axis at different orientations and with different helical faces presented toward the interaction surface, in seven distinct poses in total. For orientation, the cAD N-terminus is shown as a sphere and one helical face is colored red. See also *Figure 6—figure supplement 1E*. (**D**) Distinct binding poses were defined by clustering similar structures based on cAD backbone root mean square distance, and the number of poses seen in the 10 best-scoring models was counted for cAD interactions with the KIX domain and ABD1. See also *Figure 6—figure supplement 1E–G*. (**E**) Sidechain locations of all Leu (red), Phe (yellow), and Asp (blue) residues of all cADs interacting with the KIX and ABD1 surfaces (gray) are marked by dots. Leu and Phe distributions were similar to each other: there was no binding pocket that selectively preferred one residue over the other, despite large differences in their size and shape.

The online version of this article includes the following source data and figure supplement(s) for figure 6:

**Source data 1.** List of all Med15 ABD and core AD interactions modeled using FlexPepDock.

**Source data 2.** Structural models of Med15 ABD and core AD interactions generated by FlexPepDock.

**Figure supplement 1.** Validation and additional analysis for structural modeling of core AD interaction with Med15 ABDs.

complementarity to maximize interface area and minimize solvent-exposed area. However, the flat surface of the KIX domain was ineffective in burying hydrophobic cAD residues, which on average had as much surface area exposed to solvent as area contacting the KIX domain (*Figure 6B*, top). For example, the best-scoring Pdr1 AD model used Ile and Leu residues along one helical face to contact the KIX domain, but this left Trp and Tyr residues on the opposite face exposed to solvent (*Figure 6—figure supplement 1C*). The poor hydrophobic shape complementarity of the KIX domain was frequently mitigated by multiple salt bridges and cation-pi interactions (*Figure 6—figure supplement 1D*). ABD1, which formed fewer ionic interactions, engaged cADs through a larger surface formed by its contoured hydrophobic ridge (*Figure 6B*, bottom). Nevertheless, hydrophobic residues of most cADs were incompletely buried, with those residues showing greater than 20% of area exposed to solvent in over half of all structures. Thus, despite the central role of hydrophobic AD residues in activation and Med15 binding, hydrophobic shape complementarity is not an important feature of ABD-cAD interactions.

Consistent with weak constraints on the shape of the binding interface, all cADs bound in diverse poses. For example, the 10 best-scoring models of the War1 cAD bound the KIX domain with its helix axis at different orientations and with different helical faces presented toward the interaction surface (*Figure 6C*). To quantify this diversity, we defined the unique binding poses of each ABD-cAD interaction by clustering similar structures based on cAD backbone root mean square distance (2 Å distance cutoff), and then counted the number of poses adopted by the 10 best-scoring models. The 10 best-scoring models of nearly every cAD, when interacting with either domain, adopted six or more distinct poses (*Figure 6D*). To ensure that this did not result from insufficient sampling, we generated tenfold more candidate models (500,000 total) for six cADs binding the KIX domain. For all six cADs, the 10 best-scoring models still adopted seven or more distinct poses, despite spanning a narrower range of Rosetta scores (*Figure 6—figure supplement 1E–F*). Disordered cADs, lacking helical content, adopted especially many distinct poses; also, poses were frequently sampled only once, suggesting that disordered cADs inhabit an extremely large conformational space that may reduce the entropic cost of binding (*Figure 6—figure supplement 1G–H*). Altogether, these structures point to a shape-agnostic nature of Med15 ABD interaction surfaces and support the idea of a multi-pose, fuzzy mode of interaction.

To explore the consequence of fuzzy binding for amino acid sequence constraints, we took Leu, Phe, and Asp residues from models of all cADs and plotted their binding positions on the KIX and ABD1 interfaces (*Figure 6E*). As expected, the hydrophobic Leu and Phe residues occupied regions distinct from the negatively charged Asp residues. However, the Leu and Phe distributions were similar to each other: there was no binding pocket that selectively preferred one residue over the other, despite large differences in their size and shape. The diversity of AD sequences may thus be understood in terms of the fuzzy nature of the AD-ABD interaction, which only loosely constrains the characteristics of hydrophobic AD residues.

## The high valence of TF-Mediator interactions enables tunable, long-lived, yet dynamic binding

Pull-down experiments with individual ABDs showed no appreciable binding to any of nine ADs (*Figure 7—figure supplement 1A*), consistent with the previous suggestion that multiple ABDs are required for strong binding to Med15 (*Brzovic et al., 2011*; *Tuttle et al., 2018*). Multiple interaction sites were also common within ADs: PADDLE identified 42 ADs (28%) that contained two or more non-overlapping cADs (*Figure 7—figure supplement 1B*, *Figure 3—source data 1*, Materials and methods). We took 47 pairs of adjacent cADs and measured activation by the two cADs individually or in tandem. For 40 of the pairs, the tandem cADs activated more strongly than expected from the combined activation of the individual cADs (4.0-fold median increase, *Figure 7A*). Thus, an increased valence of interaction sites in both Med15 and ADs drives stronger binding and activation.

At larger scales, the number of interaction sites is further multiplied because some TFs contain multiple ADs, many TFs bind DNA as dimers, and many genes have several TF binding sites (*Hahn and Young, 2011*; *Spitz and Furlong, 2012*). To understand the advantages conferred by the high valence of TF-Mediator interactions, we studied the kinetics in vitro of the initial step of activation: TF-driven recruitment of Mediator to DNA. We performed these experiments with Gcn4, which forms homodimers and has an extended AD consisting of three cADs. We isolated budding yeast Mediator complex, coupled DNA containing a Gcn4 motif to a NeutrAvidin-coated surface, added

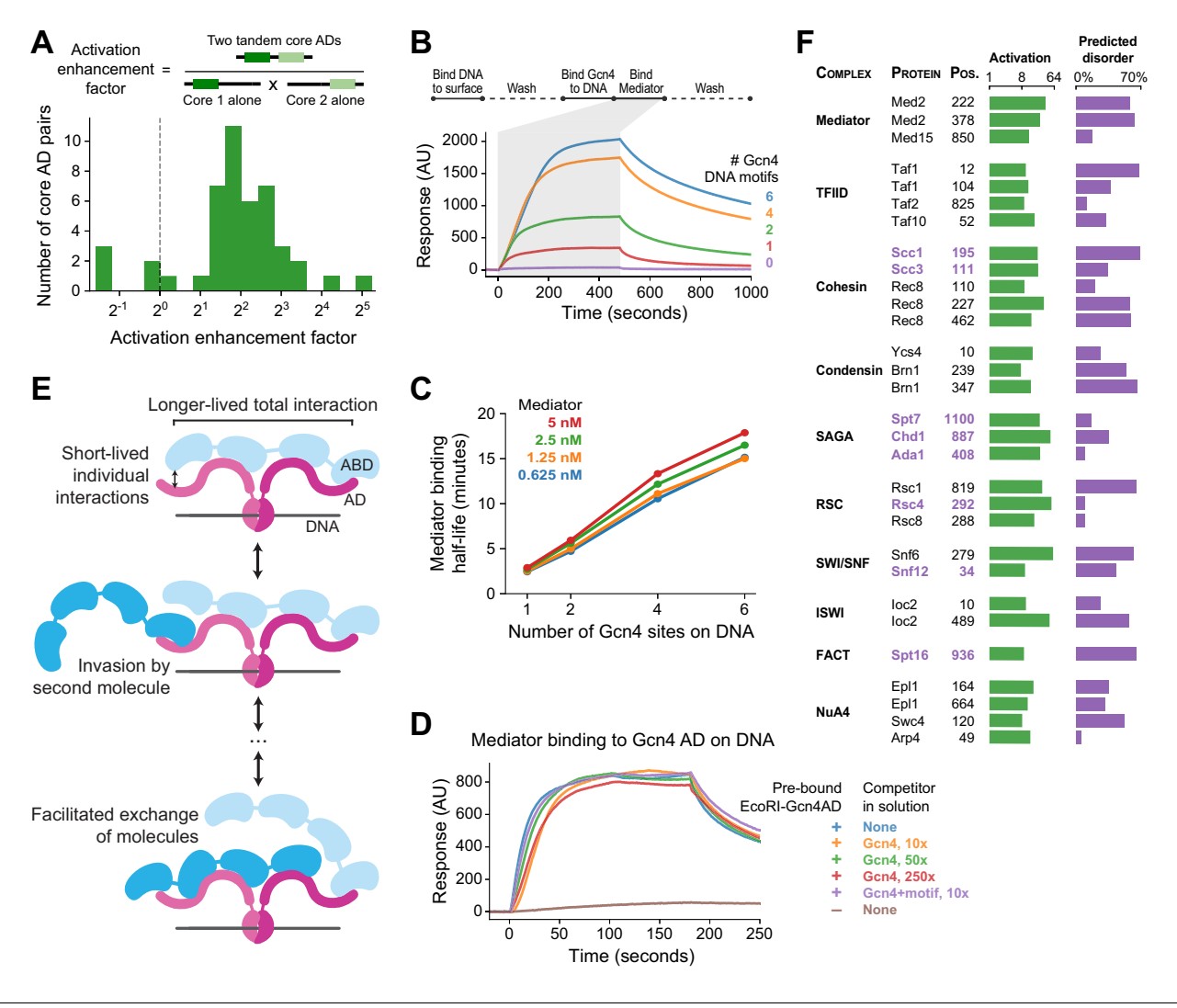

**Figure 7.** Functional consequences of high valence coactivator interactions. (**A**) We took 47 pairs of adjacent cADs and measured their activation enhancement factors—the activation of both cADs in tandem divided by the product of activation by each cAD individually. For 40 pairs, activation was enhanced when cADs were in tandem, with a median enhancement factor of 4.0-fold. (**B**) Kinetics of Mediator complex recruitment to DNA by Gcn4. (*Top*) We coupled DNA containing a Gcn4 motif to a NeutrAvidin-coated surface, added Gcn4, and measured real-time binding of Mediator by surface plasmon resonance. The Mediator-binding step also included 7.5 nM Gcn4 to maintain DNA-bound Gcn4. (*Bottom*) Real-time binding of 2.5 nM Mediator to Gcn4-DNA complexes (0 to 480 seconds) and subsequent dissociation (480 seconds onwards). DNA templates contained 0, 1, 2, 4, or 6 copies of the Gcn4 motif. AU, arbitrary units. (**C**) The interaction half-life of Mediator with Gcn4-DNA complexes was proportional to the number of Gcn4 motifs and independent of the concentration of Mediator used in the binding step. See also *Figure 7—figure supplement 1C–D*. (**D**) Gcn4 competition assay. We purified a fusion of the Gcn4 AD to nuclease deficient EcoRI(E111Q), which resides on DNA for several hours, bound it to DNA containing a single EcoRI site, and measured Mediator recruitment in the presence or absence of excess Gcn4 competitor. Mediator was at 5 nM in all six conditions. AU, arbitrary units. See also *Figure 7—figure supplement 1E*. (**E**) Model of Mediator-TF interaction that explains high-affinity but dynamic binding. Multiple weak, short-lived interactions between individual Mediator ABDs (blue) and TF ADs (maroon) together drive longer-lived high-affinity binding. However, rapid association-dissociation equilibrium of an AD allows a second Mediator molecule to interact with Mediator-bound TF, facilitating the dynamic exchange of one Mediator molecule by another. The same mechanism would allow exchange of one TF by another. (**F**) An activation screen of nuclear proteins identified ADs in all major coactivator and chromatin modifying complexes. Protein complex, protein name, and start position of the 53-aa AD is labeled, and experimentally measured activation (green) and fraction of residues predicted to be disordered (purple; in D2P2) is shown. ADs that are predominantly disordered or unresolved in PDB structures are displayed in purple text. Taf10 is also a subunit of SAGA and Arp4 is also part of the INO80 chromatin remodeling complex. See also *Figure 7—figure supplement 1F–G* and *Figure 1—source data 1*.

The online version of this article includes the following figure supplement(s) for figure 7:

**Figure supplement 1.** Additional data and analysis.

Gcn4, and measured real-time binding of Mediator by surface plasmon resonance (*Figure 7B*, *Figure 7—figure supplement 1C*). Mediator bound to the Gcn4-DNA complex with an apparent dissociation constant of $K_D$ = 14 nM ± 2 nM and an interaction half-life of 2.7 ± 0.2 min. We repeated the experiment with DNA containing 2, 4, or 6 copies of the Gcn4 motif. Mediator bound the resulting Gcn4-DNA complexes with identical on-rates. Mediator was also recruited more efficiently, with more Mediator bound per mole of Gcn4 than on templates with only one motif (*Figure 7—figure supplement 1D*). The interaction half-life, however, increased with additional copies of the Gcn4 motif—up to 16 min when six motifs were present (*Figure 7B–C*). If all copies bound Mediator independently, then the half-life should be the same, regardless of the number of copies. The presence of additional, neighboring ADs retards dissociation by enabling recapture of Mediator released from one AD through binding to another. A slower dissociation rate corresponds to an increase in apparent affinity constant.

Conversely, we found that the multiplicity of ABDs in Mediator facilitated binding of Gcn4 ADs. When the surface plasmon resonance experiment was repeated with large excess Gcn4 in solution, there was almost no change in the measured on-rate. Even 160-fold excess of Gcn4 had only minimal effect on the rate of Mediator binding and final level achieved (*Figure 7—figure supplement 1E*). To maintain stable TF-DNA complexes without TFs in solution, we repeated these experiments using a fusion of the Gcn4 AD to nuclease deficient EcoRI(E111Q), which resides on DNA for many hours and also forms homodimers (*Wright et al., 1989*). Still, Mediator with no Gcn4 competitor or with 250-fold excess of Gcn4 competitor bound similarly (*Figure 7D*). Excess competitor Gcn4-DNA complexes also failed to slow Mediator binding to the surface. Evidently, binding of Gcn4 to Mediator does not appreciably block binding of other Gcn4 molecules. This behavior may be explained by the occurrence of multiple Gcn4-binding sites on Mediator and weak interaction of a cAD with any one site. Rapid association-dissociation equilibrium of a cAD allows a second Gcn4 molecule to interact with Gcn4-bound Mediator, increasing the frequency of exchange of one Gcn4 molecule by another (*Figure 7E*). Such facilitated exchange provides a rationale for both the fuzzy nature of the AD-ABD interaction and the multiplicity of ABDs.

## Activation domains are present in all major coactivator complexes

PADDLE predictions and previous experiments (*Liu and Myers, 2015*) identified two strong ADs in the Med2 subunit of Mediator, which is adjacent to Med15 in the Mediator complex, suggesting a broader role for ADs beyond TF proteins (*Figure 7—figure supplement 1F*). We used PADDLE to search for ADs across all nuclear-localized non-TF proteins in yeast and selected 1485 fragments that showed potential for activation (Materials and methods). Upon pooled in vivo testing, we found 290 fragments across 229 proteins that activated significantly (p<0.001 by Z-test, *Figure 1—source data 1*). To see how often PADDLE fails to identify acidic ADs, we also included 291 control fragments that had extremely high acidic and hydrophobic content but were predicted not to activate. Zero control fragments activated (*Figure 7—figure supplement 1G*), showing that PADDLE likely identified all acidic ADs that exist and again demonstrating that AD function cannot be predicted from amino acid composition alone.

ADs were remarkably widespread and were found in proteins associated with diverse functions including transcriptional regulation, RNA splicing, DNA repair, ribosome biogenesis, and protein degradation. We focused on the ADs in transcription-associated proteins. Fourteen ADs were found in proteins that bind DNA and regulate pathway-specific genes; these proteins were not included in the original tiling library because they bind DNA only through a TF partner. The two predicted Med2 ADs were also confirmed to activate, 36-fold and 25-fold in our assay.

Notably, ADs were identified in all major coactivator and chromatin modifying complexes, including Mediator, TFIID, cohesin, condensin, SAGA, RSC, SWI/SNF, ISWI, NuA4, and FACT (*Figure 7F*). These complexes contained 2.9-fold more activating fragments than expected by chance (p<$10^{-12}$ compared to randomly shuffled controls), suggesting a functional role. Domains that activate on their own could be inactive in their natural context if buried within a protein or protein complex. To investigate this possibility, we cross-referenced coactivator ADs with all available structures in the Protein Data Bank (PDB) and with the Database of Disordered Protein Predictions (D2P2) (*Berman et al., 2000*; *Oates et al., 2013*). In fact, 23 ADs were predicted in D2P2 to be highly disordered (more than 30%) or were predominantly disordered or unresolved in PDB structures (*Figure 7F*), and so may be exposed and active in vivo. For example, cohesin, which binds Mediator

and forms enhancer-promoter contacts through loop extrusion (*Davidson et al., 2019*; *Kagey et al., 2010*; *Kim et al., 2019*; *Sanborn et al., 2015*), contained strong and likely-disordered ADs in its subunits Scc1 (RAD21 homolog), Scc3 (STAG1/2 homolog), and Smc3, as well as multiple ADs on the meiotic-specific subunit Rec8. These cohesin ADs, as well as ADs within other coactivators, may play a role in gene regulation by driving interactions with Mediator.

## Discussion

Although activation domains (ADs) have been studied for over three decades, their locations in TFs and mechanisms of activation have remained enigmatic. Combining quantitative, high-throughput experiments and computational modeling, we determined the biophysical principles underlying activation by acidic ADs. We trained a neural network to predict ADs in yeast and humans, which led to the discovery of new ADs and activation mechanisms. While an amphipathic helix can enhance activation, most ADs simply activate through an abundance of clustered acidic and bulky hydrophobic residues. ADs bind the hydrophobic and basic surface of Med15 activator-binding domains (ABDs) through a fuzzy interface. The low shape complementarity of this interaction only weakly constrains AD sequences, explaining their high diversity. We show that the dynamic nature of transcriptional signaling arises from the fuzzy and multivalent nature of TF-Mediator interactions. Finally, an expanded role for ADs is suggested by our finding that ADs are present in all major coactivator complexes.

### Quantitative, high-throughput measurements of activation enabled prediction of new ADs and revealed the principles underlying acidic activation

We obtained high-throughput measurements of activation strength that were both quantitative and precise. This enabled us to detect—and predict with PADDLE—how activation strength depends on amino acid sequence, amino acid composition, and valence. Quantitation was achieved by inducing artificial TF (aTF) binding for a brief, defined period so that GFP levels were representative of transcriptional activation (*McIsaac et al., 2013*), and by partitioning the range of GFP signal into eight levels for sorting. Others have measured activation only as an on-off binary system or screened for non-quantitative measures of activation, such as cell survival or pull-down of a cell surface marker (*Arnold et al., 2018*; *Ravarani et al., 2018*; *Tycko et al., 2020*). We achieved high precision by restricting measurements to cells with a defined level of aTF expression. Controlling for this frequently overlooked confounding variable was critical because activation was strongly dependent on aTF abundance, especially at low levels of expression (*Figure 1—figure supplement 1E*). Another approach is to account for TF expression by sorting based on the GFP reporter signal divided by aTF expression (*Staller et al., 2018*; *Staller et al., 2021*); this introduces substantial variability, because activation has a non-linear relationship with aTF expression (*Figure 1—figure supplement 1E*), and because this ratio can be systematically biased by features that affect aTF expression (*Figure 1—figure supplement 1F*). This may be why it has been concluded that acidic residues are not necessary for Gcn4 activation, whereas using a similar reporter system we found definitive evidence to the contrary.

By this approach, we found that all ADs in *S. cerevisiae* employ a shared acidic and hydrophobic basis and trained a neural network termed PADDLE to predict sequences that activate on this basis. PADDLE predicted ADs in human TFs with 92% accuracy, indicative of broad conservation of the acidic activation mechanism, and identified hundreds of new ADs in human TFs and virus proteins. PADDLE could further be used to interpret the functional impact of cancer-associated and naturally occurring mutations in human ADs. Predicted ADs from multiple species and instructions for running PADDLE are available online at paddle.stanford.edu.

During our development of PADDLE, a predictor of yeast activation named 'ADpred'' was published (*Erijman et al., 2020b*). ADpred, a shallow neural network trained on random protein sequences, achieved a prediction accuracy of 93% on random sequences. However, its accuracy on wild-type sequences was much lower: only 33% of ADs predicted by ADpred activated in our experiments, and 29% of the ADs we identified were not predicted by ADpred, only 3.1-fold and 2.3-fold better than random predictions, respectively (alternatively, tile-wise AUPRC of 0.294; *Figure 3—figure supplement 1J–K*, Materials and methods). This discrepancy may be because the random

sequences on which ADpred was trained do not sufficiently represent the vastly larger space of actual protein sequences; for example, random sequences rarely form significant secondary structure. This failure to generalize highlights the importance of matching neural network training data to the prediction task and underscores the advantage of experiments on wild-type and related mutant sequences.

PADDLE predictions were also crucial for generating, refining, and testing hypotheses for activation mechanisms. For example, we unexpectedly discovered that hydrophobic residues could inhibit activation by noticing that sub-domains of certain non-activating regions were predicted to activate on their own. More generally, predictions on AD subsequences identified the core domains responsible for activation at single-amino acid resolution, focusing subsequent experiments on these domains. By combining a high-throughput assay to measure and a machine learning algorithm to predict activity of protein sequences, we have demonstrated a design-build-test-learn cycle that could be applied to accelerate discovery of protein function in other areas of research.

This approach yielded the most detailed view to date of the principles governing acidic AD sequence, which we summarize as follows. Activation arises from an abundance of acidic and bulky hydrophobic residues, especially Asp, Trp, Phe, and Leu. The most potent activators cluster these residues densely, forming short core ADs. In most ADs, and especially with high hydrophobic content or abundant Phe and Trp residues, this clustering is sufficient to activate in a composition-driven manner, regardless of sequence or secondary structure. ADs with lower hydrophobic content, particularly those with many Leu residues, instead must position their hydrophobic residues along one face of an alpha helix to activate. Overall, the loose constraints on sequence explain the remarkable diversity and lack of evolutionary conservation of ADs.

The unusual plasticity in AD sequence has several advantages. It facilitates spontaneous evolution of new ADs, since an appreciable proportion of even random sequences can activate. This flexibility would allow organisms to develop new transcriptional circuits responsive to their unique needs. Furthermore, in contrast to classical interactions in which structure and function are dependent on a few key residues, activation strength can be adjusted gradually by mutation. Indeed, the observation that nearly all core ADs could increase their strength with simple mutations demonstrates that proper regulation requires precisely adjusted rather than maximized activation. Similarly, AD sequences can preserve their activity while evolving other features, such as in the Gal4 C-terminal AD, whose sequence is specifically bound and inhibited by Gal80 in the absence of galactose (*Johnston et al., 1987*; *Ma and Ptashne, 1987*).

## ADs bind Mediator using a shape-agnostic fuzzy interaction, which explains the heterogeneity of AD sequences and enables facilitated exchange of bound molecules

With the use of mRNA display, we found that 73% of ADs bound the Med15 subunit of Mediator and that binding strength was strongly predictive of activation for thousands of wild-type and mutant protein sequences. In contrast, a five-protein TFIID subcomplex bound only 18% of ADs, all of which also bound Mediator. We conclude that Mediator recruitment by TFs is a key driver for gene activation, and Mediator recruitment is largely determined by affinity of ADs for the Med15 subunit. These findings are in keeping with reports that Mediator recruitment occurs independently of other coactivators and can stimulate the recruitment of other coactivators (*Ansari and Morse, 2013*; *Ansari et al., 2014*; *Govind et al., 2005*).

How is Med15, through its four activator-binding domains (ABDs), able to interact with such a large diversity of AD sequences? Despite the importance of hydrophobic interactions, our structural modeling showed that neither the Med15 KIX domain nor ABD1 provided enough shape complementarity to bury the hydrophobic residues of cADs. Consistent with this, most cADs did not need to form an alpha helix to activate. The lack of shape constraint has two important consequences. First, all modeled cADs bound each ABD in many distinct, equally favored poses. We therefore suggest that fuzzy binding to Med15, previously demonstrated for the Gcn4 AD, is employed by all ADs. Second, neither ABD showed favored binding positions of Leu versus Phe residues of the cAD, suggesting that the shapes of those residues were unimportant. Thus, the loose constraints on binding shape explain the loose constraints on AD sequence. Instead, each ABD can be approximated as a hydrophobic surface, which engages hydrophobic AD residues through simple hydrophobic forces, flanked by basic residues, which form salt bridges and cation-pi interactions with acidic and aromatic

AD residues. Clusters of hydrophobic residues adjacent to ADs inhibit activation, presumably because they compete for interaction with hydrophobic AD residues.

Many experiments have shown that transcriptional initiation involves a cycle in which (1) TFs recruit Mediator to enhancers or upstream activating sites, (2) Mediator contacts the promoter and orchestrates a series of events starting with chromatin rearrangement, followed by pre-initiation complex formation and finally RNA polymerase II recruitment, and (3) Mediator facilitates phosphorylation of the polymerase C-terminal domain, which releases polymerase into transcriptional elongation and triggers dissociation of Mediator (*Jeronimo and Robert, 2014*; *Knoll et al., 2018*; *Robinson et al., 2016*; *Whyte et al., 2013*; *Wong et al., 2014*). Mediator release is apparently necessary because stably recruiting Mediator by fusing individual subunits to a DNA-binding domain failed to activate in nearly all subunits tested (*Wang et al., 2010*). The cycling of Mediator complexes increases the responsiveness of transcriptional regulation in at least two ways. First, it requires certain steps to be repeated in order to continue initiating transcription, preventing the system from locking into an activated state. Second, it frees Mediator complexes to participate in regulation of other genes, which all depend upon and compete for a relatively limited supply of Mediator (*Flanagan et al., 1991*; *Gill and Ptashne, 1988*). Thus, TFs must recruit Mediator in a manner that is specific and high-affinity but still dynamic.

Our kinetic experiments showed that Mediator binds Gcn4-DNA complexes with high affinity but that excess Gcn4 does not impede this interaction, showing that interacting Mediator and Gcn4 molecules can exchange rapidly. If a bimolecular interaction occurs through a single site, a competing molecule cannot bind until the first molecule dissociates which, in a case of high affinity interaction, is a slow process. Our observations are indicative of facilitated exchange, arising from the multiple sites and fuzzy nature of the Gcn4-Mediator interaction (*Figure 7E*). Fuzzy interactions allow for rapid association and dissociation, because a high proportion of random encounters lead to weak but productive binding (*Ferreira et al., 2005*; *Sugase et al., 2007*; *Tompa and Fuxreiter, 2008*). One among multiple binding sites will frequently be available for a second molecule, which can invade and facilitate release of the first. Similarly, accelerated dissociation or exchange of the *E. coli* sequence-flexible DNA-binding protein Fis in the presence of competitor proteins was proposed to occur by transitioning through a destabilizing Fis-DNA-Fis ternary complex (*Graham et al., 2011*; *Kamar et al., 2017*). These advantageous kinetics provide a rationale for the prevalence of multivalent fuzzy interactions among transcription proteins.

## Coactivator complexes contain ADs that may drive cross-interaction

ADs have primarily been characterized in TFs where they serve to recruit coactivators such as Mediator, TFIID, SAGA, SWI/SNF, and NuA4 (*Hahn and Young, 2011*; *Mitchell and Tjian, 1989*). Our finding that functional ADs are also present in all coactivator complexes suggests that ADs have broader roles. First, coactivator ADs could interact with ABDs within the same complex, limiting their weak or non-specific binding to TFs. These auto-inhibited coactivators could still bind to ADs clustered on DNA through facilitated exchange. For example, we found that two cADs activated more efficiently and two DNA-bound Gcn4 dimers recruited Mediator more efficiently in tandem than individually. Second, coactivator ADs could amplify activation by mediating interactions between coactivators (*Ansari and Morse, 2013*; *Liu and Myers, 2015*). For example, upon Mediator binding to TFs, the two ADs on Med2 would be liberated from auto-inhibition and could recruit other coactivators. Alternatively, Mediator and other coactivators could cluster in the nucleoplasm through AD-ABD cross interactions and bind together to promoters. Either mode of interaction would explain why recruitment of the Med2/Med3/Med15 subcomplex suffices to recruit SWI/SNF to the *CHA1* promoter and suffices for activation of the *ARG1* gene, while deletion of both Med2 ADs impairs transcription of galactose-inducible genes (*Ansari et al., 2014*; *Liu and Myers, 2015*; *Zhang et al., 2004*).

Phase separation of Mediator and TFs has recently been demonstrated in vitro and is proposed to underlie enhancer-promoter clustering and transcriptional activation in vivo (*Boija et al., 2018*; *Chong et al., 2018*; *Sabari et al., 2018*; *Shrinivas et al., 2019*). Phase separation may occur when a protein makes dynamic self-interactions through two or more binding domains, (*Banani et al., 2016*). Interactions between the four Med15 ABDs and the two Med2 ADs make possible phase separation of the Mediator tail subcomplex; other coactivators, with both ADs and ABDs, might phase separate as well. The functional role of such phase separation, if it occurs, is unclear. A parsimonious

interpretation suggests that phase separation of coactivators and TFs is simply a macroscopic consequence of the high valence and dynamism of coactivator-TF interactions, features which may serve an altogether different purpose.

### Solving the enigma of AD structure and function

ADs have long been enigmatic, due to the apparent mismatch of their structure and function: AD sequences are abundant among random polypeptides, and yet ADs bind their targets with high specificity; AD peptides are apparently disordered, and yet they bind with high affinity. We now understand that these structural features of ADs are ideally suited to their functions—they render AD-target interaction dynamic, through rapid yet weak binding to single sites and strong binding yet rapid displacement from multiple sites. Dynamic AD-target interaction may serve various purposes, such as a rapid response to changing conditions, and the recruitment of multiple AD-bearing proteins to a single target. For example, Mediator bound to RNA polymerase II at a promoter might interact transiently with TFIID, SAGA, SWI/SNF complex and other proteins during transcription. This mechanism may be employed with other unstructured sequences and high valence targets in other cellular processes.

# Materials and methods

**Key resources table**

| Reagent type (species) or resource | Designation | Source or reference | Identifiers | Additional information |
|---|---|---|---|---|
| Cell line (*S. cerevisiae*) | BY4711 | ATCC | 200873 | *MATalpha trp1delta63* |
| Cell line (human) | HEK293T | ATCC | CRL-3216 | |
| Sequence-based reagent | Oligo pool libraries | Twist Bioscience | Custom synthesis | Sequences provided in source data files |
| Recombinant DNA reagent | pMVS142-pACT1-mCherry-Zif268-EBD-MCS-KAN | Addgene | 99049 | Barak Cohen |
| Recombinant DNA reagent | pMVS102-P3-GFP-NATMX6 | Addgene | 99048 | Barak Cohen |
| Chemical compound, drug | Fetal bovine serum | Millipore | TMS-031-B | |
| Strain, strain background (*Escherichia coli*) | XL10 Gold Ultracompetent Cells | Agilent | 200315 | For cloning |
| Peptide, recombinant protein | Phusion polymerase | New England Biolabs | M0531L | |
| Commercial assay or kit | AMPure XP beads | Beckman Coulter | A63880 | |
| Recombinant DNA reagent | pFN26A BIND plasmid | Promega | E1380 | |
| Recombinant DNA reagent | pGL4.35 plasmid | Promega | E1370 | |
| Commercial assay or kit | Dual-Glo Luciferase assay | Promega | E2920 | |
| Recombinant DNA reagent | pBirAcm | Avidity | AVB99 | |
| Strain, strain background (*Escherichia coli*) | BL21 Star (DE3) competent cells | ThermoFisher | C601003 | For protein expression |
| Commercial assay or kit | Ni-NTA Suerflow resin | Qiagen | 30410 | |
| Recombinant DNA reagent | pFASTBac1 plasmid | ThermoFisher | 10360014 | |
| Commercial assay or kit | MEGAscript T7 transcription kit | ThermoFisher | AM1333 | |
| Peptide, recombinant protein | T4 DNA ligase | New England Biolabs | M0202S | |
| Other | Amicon Ultracel 0.5 mL 30K MWCO column | Millipore Sigma | UFC503024 | |
| Commercial assay or kit | Model 422 Electro-Eluter | Bio Rad | 1652976 | |
| Commercial assay or kit | Retic Lysate IVT Kit | ThermoFisher | AM1200 | |

*Continued on next page*

*Continued*

| Reagent type (species) or resource | Designation | Source or reference | Identifiers | Additional information |
|---|---|---|---|---|
| Sequence-based reagent | PF30P oligo | IDT | Custom synthesis | /5Phos/AA AAA AAA AAA AAA AAA AAA A/iSp9//iSp9//iSp9/AC C/3Puro/ |
| Peptide, recombinant protein | SuperScript II Reverse Transcriptase | ThermoFisher | 18064014 | |
| Commercial assay or kit | Dynabeads MyOne Streptavidin T1 | ThermoFisher | 65602 | |
| Sequence-based reagent | Salmon Sperm DNA | ThermoFisher | 15632011 | |
| Peptide, recombinant protein | BSA | New England Biolabs | B9000S | |
| Software, algorithm | PADDLE | This paper | github.com/asanborn/PADDLE | |

## Resource availability

Further information and requests for resources and reagents should be directed to and will be fulfilled by the Lead Contact, Roger Kornberg (kornberg@stanford.edu). Raw sequencing data for in vivo activation and in vitro binding assays have been deposited in the Gene Expression Omnibus (GEO) database, accession number GSE173156.

## Experimental model

The *S. cerevisiae* strain BY4711 (ATCC 200873) with genotype *MATalpha trp1delta63* was used for activation experiments. Standard cell growth conditions were 30°C incubation in YPD medium (1% w/v yeast extract, 2% w/v peptone, 2% w/v dextrose) or synthetic complete (SC) medium (*Dunham, 2015*). Growth in SC medium lacking tryptophan (SC-Trp) was used to select for transformants with the pRS414 plasmid backbone. Selection for stable transformants harboring the *natMX6* marker gene was performed by supplementing media with 100 µg/mL nourseothricin (NAT). Selection for stable transformants harboring the *KanMX* marker gene was performed by supplementing media with 200 µg/mL geneticin (G418). In activation assays, overnight cultures were diluted to OD600 = 0.4 in SC-Trp-NAT-G418 with 10 nM beta-estradiol and grown for 4 hr at 30°C before fluorescence-activated cell sorting (FACS).

HEK293T cells (ATCC CRL-3216) were cultured in DMEM with high glucose and GlutaMAX (Thermo 10566016) supplemented with 10% fetal bovine serum (Millipore TMS-013-B) at 37°C in 5% $CO_2$. Yeast and human cells were used directly from ATCC, which authenticates all cell lines using STR profiling and tests for mycoplasma contamination.

## Activation screen plasmid library construction

The artificial TF (aTF) expression cassette, which contains a *KanMX* gene and an *ACT1* promoter driving expression of an aTF (comprising an mCherry tag, mouse Zif269 DBD, and estrogen binding domain) with a multiple cloning site for insertion of protein fragments of interest, was derived from pMVS142_pACT1_mCherry_Zif268_EBD_MCS_KAN, which was a gift from Barak Cohen (Addgene plasmid # 99049) (*Staller et al., 2018*). This cassette was cloned into the pRS414 backbone, which contains a *TRP1* marker (*Sikorski and Hieter, 1989*). pRS414 plasmid was digested with EcoRI-HF and SacI, the aTF cassette was PCR-amplified using Phusion polymerase (NEB), and DNA fragments were assembled through Gibson assembly and transformed into XL10-Gold Ultracompetent Cells (Agilent). The final construct, called pRS414-TFchassis, was confirmed by Sanger sequencing.

Oligo pools encoding all the protein fragments to be tested in the screen were synthesized at 12K scale (Twist Bioscience). Each oligo was 200 nt in length and contained a 20-nt 5' constant region and a 21-nt 3' constant region which were used for PCR amplification. Different sub-libraries had different 3' constant regions, which allowed specific amplification of the sub-library from the oligo pool. To enable cloning of the oligo pool into pRS414-TFchassis, 20 bp Gibson homology arms matching pRS414-TFchassis were added using PCR. To amplify each sub-library, 2.5 ng of oligo template was PCR-amplified for nine cycles using Phusion polymerase, followed by column cleanup. pRS414-TFchassis was digested with NheI-HF and AscI, the amplified oligo pool was cloned in by

Gibson assembly, and the product was transformed in two replicates, each into 100 µL of Agilent XL10-Gold cells with 5 µL of Gibson mix. Transformed cells were plated on lysogeny broth (LB) plates with Carbenicillin (Carb) to estimate efficiency and also grown in 50 mL LB Carb overnight at 37˚C for next-day plasmid extraction using Qiagen Midiprep kit.

To generate positive control plasmids for the activation screen, IDT gBlocks containing genes expressing 53-aa regions from VP16, Gcn4, and Pho4 ADs were also cloned in the same manner as above. The uncloned pRS414-TFchassis, which lacks an introduced insert, was used as a negative control.

The GFP reporter construct pMVS102_P3-GFP-NATMX6 was a gift from Barak Cohen (Addgene plasmid # 99048) (*Staller et al., 2018*). To construct a yeast GFP reporter cell line, a cassette from pMVS102_P3-GFP-NATMX6 comprising a *natMX6* marker and a P3 promoter (a modified *GAL1* promoter with six copies of the Zif268 binding site) that drives expression of a fast maturing EGFP variant (*McIsaac et al., 2013*), was PCR-amplified using Phusion polymerase. PCR primers were designed to add 40 bp overhangs with homology to the dubious ORF *YBR032W*. The PCR product was transformed into yeast using a high-efficiency method (*Gietz et al., 1992*). After overnight outgrowth at 30˚C, transformed cells were plated onto YPD-NAT, colonies were picked, and integration into *YBR032W* was confirmed by PCR amplification of the integrant and Sanger sequencing.

## Multiple plasmid transformant outgrowth assay

The in vivo activation screen requires a yeast library in which each cell contains one pRS414-TFchassis plasmid with a unique insert. The plasmid is maintained at single-copy or low levels due to its CEN/ARS origin of replication, so if a cell receives multiple different vectors upon transformation then repeated cell divisions will eventually separate the distinct vectors (*Scanlon et al., 2009*). To ensure generation of a cellular library in which a unique pRS414-TFchassis plasmid was present in each cell, an assay was developed to quantify the fraction of cells initially receiving multiple plasmids and the time required for cells to reach single-copy plasmid levels by repeated cell divisions. First, using a high efficiency transformation protocol (*Gietz et al., 1992*), approximately 50 million yeast cells in a volume of 360 µL were transformed with 2 µg total plasmid DNA, consisting of an equal mixture of aTF plasmids expressing one AD from VP16, Gcn4, or Pho4. Transformed cells were sparsely plated on SC-Trp-NAT plates either immediately or after a 24 hr or 48 hr outgrowth in SC-Trp-NAT liquid media. For each plating condition, colonies were picked for DNA extraction, followed by qPCR to measure the abundance of VP16, Gcn4, and Pho4 plasmids. After 0 hr, 24 hr, and 48 hr of growth, 2 (of 15), 2 (of 15), and 1 (of 24) colonies had multiple vectors present, respectively. Because this method cannot detect multiple transformation events of the same plasmid sequence in the same cell (e.g. double VP16 transformation), a correction was applied, yielding multiple vector rates of 20%, 20%, and 6% for each outgrowth condition, respectively.

## Pooled activation assay in yeast

The cloned pRS414-TFchassis plasmid library was transformed into the GFP reporter yeast strain using the same method as in the multiple transformant outgrowth assay above but scaled up 2.5-fold. To generate clonal control strains, cells were also separately transformed with a positive control plasmid expressing a VP16, Gcn4, or Pho4 AD or with a negative control plasmid, the pRS414-TFchassis plasmid without a protein fragment insert. Transformed cells were grown in 200 mL SC-Trp-NAT-G418 for 6 days (first library) or 5 days (second library); cells were diluted down to OD600 = 0.05 every day upon reaching saturation. After this period, cells were diluted to OD600 = 0.4 in SC-Trp-NAT-G418 containing 10 nM beta-estradiol and grown at 30˚C for 4 hr. Cells were then pelleted and washed once in FACS buffer (PBS, 2 mM EDTA, 0.1% BSA; sterile filtered), and finally suspended in FACS buffer at roughly 40 million cells/mL.

Cells were analyzed and sorted on a BD Influx Cell Sorter at the Stanford Shared FACS Facility. Cells from different sub-libraries were pooled just prior to sorting. The 10–15% largest cells were excluded from sorting, and then an additional filter was applied to limit mCherry signal to within a twofold range. For the remaining cells, the full range of GFP signal was divided into eight evenly spaced bins in log scale; cells were first sorted into the four low bins and then sorted into the four high bins. In total, 2.85 million cells and 2.65 million cells were sorted for the first and second libraries, respectively.

To define the background GFP signal, cells containing the negative control plasmid were also induced with 10 nM beta-estradiol and their distribution across the eight GFP bins defined above was measured. We found that these cells had slightly increased levels of GFP signal relative to untransformed cells, potentially because binding of the aTF alone slightly opens up chromatin around the GFP promoter. Cells expressing just one plasmid containing the VP16, Gcn4, or Pho4 AD were also individually analyzed (by similarly measuring their distribution across same eight GFP bins) and compared to results obtained for identical sequences in the pooled activation screen.

Cells sorted into each of the 8 GFP bins and a sample of total input cells (nine samples total) were grown overnight and then pelleted. Pellets were treated with lyticase and then plasmid DNA was extracted using a Qiagen Miniprep kit. The region of the plasmid encoding the variable protein fragment was PCR-amplified for 11–20 cycles (based on the OD600 of the culture) using Phusion polymerase with PCR primers that added flanking sequences corresponding to the standard Illumina sequencing primers. After a column cleanup, a second PCR was performed with primers that added sample-specific barcodes and the P5 and P7 sequences for Illumina sequencing. This second PCR product was SPRI size-selected using AMPure XP beads (Beckman Coulter) at 1x dilution, and the different samples were pooled for paired-end sequencing on the Illumina NextSeq 550. Roughly 2 million reads of $2 \times 76$ bp length were sequenced per sample.

## Activation assay in human cells

The pFN26A BIND plasmid (Promega), which contains a Renilla luciferase gene used for normalization, was used to express domains of interest fused to a GAL4 DBD. Genes expressing 50 randomly chosen putative human ADs predicted by PADDLE were synthesized in a single oPool (IDT) with Gibson homology arms, amplified using Phusion polymerase, and cloned into pFN26A BIND (digested with AsiSI and PmeI) using Gibson assembly. The product was transformed into Agilent XL10 Gold cells, and then plated onto LB Carb plates. Forty-three colonies were picked, which yielded clones of 25 correct unique ADs. In addition, plasmids expressing a VP16 AD-positive control or three random protein sequences for negative controls were individually cloned in the same manner.

HEK293T cells were plated at 24,000 cells per well of a 96-well plate. The next day, cells were transfected using Lipofectamine 3000 with 50 ng of cloned pFN26A BIND plasmid and 50 ng of pGL4.35 (Promega), which contains a Firefly luciferase reporter gene under the control of a minimal promoter driven by nine upstream GAL4 DNA-binding sites. Each plasmid was transfected into triplicate wells. Seventy-two hours after transfection, luminescence from renilla and firefly luciferases were measured using the Dual-Glo Luciferase assay system (Promega) on a BioTek Synergy two plate reader. Measured activity of each predicted AD or control sequence was computed by dividing the Firefly luciferase signal by the Renilla signal and then averaging the triplicate values. Fold-activation was computed by dividing the activity of each predicted AD by the mean activity of the three negative controls, Z-scores were computed based on log-scale fold-activation by standardizing the negative controls to mean of 0 and variance of 1, and p-values were computed with one tailed Z-tests.

## Cloning and purification of proteins
### Med15

*S. cerevisiae* Med15 contains four ABDs: KIX (residues 6–91), ABD1 (residues 158–238), ABD2 (residues 277–368), and ABD3 (residues 484–651) (*Herbig et al., 2010*). We purified the N-terminal portion of Med15 containing the four ABDs, termed Med15K123 and consisting of wild-type residues 1–238, 273–372, and 484–651, as well as each of the four individual Med15 ABDs. Each was cloned and purified with a C-terminal 6xHis tag and a N-terminal Glutathione S-transferase tag, TEV cleavage site, and AviTag (e.g. GST-TEV-Avi-Med15K123-6xHis). DNA expressing Med15K123 and the GST-TEV-Avi tag fusion was codon optimized, synthesized as gBlocks from IDT, and cloned into a pET-28a vector digested with NotI-HF and NcoI-HF (NEB) using Gibson assembly. Individual ABDs were also PCR-amplified from the Med15K123 gBlock and then cloned similarly. Expression plasmids, together with pBirAcm (pACYC184, Avidity), which expresses BirA enzyme for in vivo biotinylation, were transformed into BL21 Star (DE3) cells (Thermo) and cultured in LB media with kanamycin and chloramphenicol, supplemented with 50 µM biotin. Expression was induced by addition of 0.5 mM IPTG, and cells were grown overnight at 16°C. Cell pellets were suspended in wash buffer (50 mM Tris pH 8.0, 100 mM NaCl, 1 mM EDTA), pelleted, and then resuspended in lysis buffer (50 mM

HEPES pH 7.0 at 4°C, 500 mM NaCl, 10% glycerol, 40 mM imidazole, 1 mM DTT, 0.2 mM PMSF, 2 mM benzamidine, 1 mg/mL pepstatin A, 2.5 mg/mL leupeptin). Cells were lysed by passaging twice through the Avestin Emulsiflex C3 (ATA Scientific) at 15,000 psi. The extract was clarified by centrifugation at 40,000 rpm in a T647.5 rotor for 30 min. Clarified extract was bound to Ni-NTA Superflow resin (Qiagen) for 30 min at 4°C with rotation, washed with 20 column volumes of lysis buffer, and eluted in two column volumes of elution buffer (composed of lysis buffer with 500 mM imidazole). Further enrichment of the full-length purified protein was achieved by binding to streptavidin beads and washing three times before use in pull-down experiments.

## TFIID subcomplex

The open-reading frames of *TAF4*, *TAF5*, *TAF6*, *TAF9*, and *TAF12* were PCR-amplified from *S. cerevisiae* genomic DNA (BY4741) and individually cloned into pFASTBac1 (ThermoFisher). *TAF12* was further modified to include an N-terminal 2xProtA tag followed by an AviTag for biotinylation. These tags were separated by a TEV protease cleavage site. BirA, derived from pBirAcm (Avidity), was also PCR-amplified and cloned into pFASTBac1. All plasmids constructs were sequence verified. For protein expression, recombinant baculoviruses were generated per manufacturer's instruction using the Bac-to-Bac expression system (ThermoFisher). The 5-Taf complex, composed of Taf4, Taf5, Taf6, Taf9 and 2xProtA-TEV-Avi-Taf12, was expressed via viral co-infection of Sf9 cells (Expression Systems) with recombinant baculovirus, using an estimated MOI of 5. Taf12 was biotinylated in vivo, also via co-infection with a recombinant baculovirus containing BirA. For expression, Sf9 cells were grown at 28°C in fernbach flasks shaking at 110 rpm in ESF921 (Expression Systems) supplemented with 50 mM D-biotin for 52–55 hr. Cell pellets were resuspended in lysis buffer (100 mM Tris-Acetate pH 7.9, 5 mM $MgSO_4$, 1 mM EDTA, 195 mM $(NH_4)_2SO_4$, 10% glycerol, 2 mM DTT, 0.2 mM PMSF, 2 mM benzamidine, 1 mg/mL pepstatin A and 2 mg/mL leupeptin). Cells were processed for purification immediately or frozen in liquid nitrogen and stored at −80°C. For purification, the resuspended cells were lysed by sonication, followed by centrifugation at 175,000 x g for 1 hr. The clarified extract was passed over IgG Sepharose, in column format, and washed with five column volumes of lysis buffer and five column volumes of wash buffer (20 mM HEPES-KOH pH 7.9, 75 mM ammonium sulfate, 1 mM $MgSO_4$, 1 mM DTT). The protein complex was eluted with 1.2 column volumes of TEV cleavage buffer (wash buffer supplemented with 15 μg/mL TEV protease) for 12–16 hr at 4°C. The TEV elution was subsequently concentrated using a Vivaspin 20 column (PES membrane, 100 kDa molecular weight cut-off) and applied to a Superose 6 Increase gel filtration column, normalized with wash buffer +5% glycerol. Peak fractions were pooled, concentrated to 4.26 mg/mL and flash frozen in liquid nitrogen.

## Mediator complex and GCN4

*S. cerevisiae* Mediator complex used in surface plasmon resonance experiments was purified as in *Robinson et al., 2012*, except that DNA removal was achieved by adding polyethylemine (PEI) to the cell extract to a final concentration of 0.2%, followed by ammonium sulfate precipitation. Gcn4 was expressed in a pET30a vector and purified also as described in *Robinson et al., 2012*.

## Gcn4AD-EcoRI(E111Q)

Residues 1–224 of Gcn4, corresponding to the full protein except the DNA-binding domain, was PCR-amplified from *S. cerevisiae* genomic DNA. The gene for a nuclease-deficient EcoRI with mutation E111Q was synthesized as a gBlock from IDT and then PCR amplified (*Wright et al., 1989*). Both PCR products were cloned into a pET-28a plasmid that was NotI-HF and NcoI-HF digested using Gibson assembly. Gcn4AD-EcoRI(E111Q)−6xHis was purified in the same manner as the Med15 proteins, except using a different lysis buffer (50 mM Tris pH 7.5, 500 mM NaCl, 10 mM beta-mercaptoethanol (bME), 20 mM imidazole, 5% glycerol, protease inhibitors) and different elution buffer (20 mM Tris pH 7.5, 300 mM NaCl, 0.1% triton X-100, 10 mM bME, 300 mM imidazole, 5% glycerol).

## Expression of mRNA display libraries

An mRNA display library was made for each of sub-library A, sub-library B, and six pooled controls (three libraries total), using the following method. Sub-libraries of wild-type and mutant TF

fragments were expressed as mRNA-tagged peptides using mRNA display with in vitro translation as described in *Takahashi and Roberts, 2009*. Specifically, sub-library pools were PCR-amplified using primers that added an upstream T7 promoter sequence and a downstream linker and HA-tag but no stop codon. Thus, all encoded peptides were of the form MSGT[variable protein sequence] TSVGGSGSYPYDVPDYA and fused to mRNA at the C-terminus. In addition to the two sub-libraries, similar DNA sequences encoding positive control ADs from VP16, Gcn4, and Pho4 and three negative control random protein sequences were synthesized as gBlocks (IDT) (containing an upstream T7 promoter sequence and a downstream linker and HA-tag but no stop codon).

All subsequent work was performed in RNase-free conditions. mRNA was in vitro transcribed for 4 hr at 37°C using the MEGAscript T7 transcription kit (Thermo) and purified with RNeasy mini columns (Qiagen). The PF30P oligo containing a poly-A sequence, spacers, and puromycin was synthesized by IDT and ligated to the 3' end of the mRNA using Moore-sharp splinted ligation by T4 DNA ligase. The ligation product was desalted by three rounds of concentrating in 0.5 mL Amicon Ultracel 30K MWCO columns and diluting in water. The desalted product was run on a denaturing 4% polyacrylamide gel with 7 M urea. The band corresponding to full-length ligated mRNA was excised from the gel and recovered using the Model 422 Electro-Eluter with 3500 MWCO caps (Bio Rad). After concentrating and desalting in the same manner, the ligated and purified mRNA was used in an in vitro translation reaction using the Retic Lysate IVT kit (Thermo) for 60 min at 30°C. MgCl$_2$ and KCl were added to final concentrations of 60 mM and 500 mM, respectively, and the sample was incubated for 15 min at room temperature to encourage puromycin fusion. The sample was then diluted in binding buffer (20 mM Tris pH 8.0, 100 mM KCl, 1 mM MgCl$_2$, 0.01% Tween-20, 0.1 mg/ mL BSA, 0.1 mM DTT) and bound to Anti-HA magnetic beads (Pierce) for 30 min at room temperature with mixing. After washing three times for 5 min with binding buffer without BSA, peptides were eluted by incubating in 1x SuperScript II buffer (50 mM Tris-HCl pH 8.3, 75 mM KCl, 3 mM MgCl$_2$; Thermo) with 0.01% Tween-20 and 0.5 mg/mL synthetic HA peptide (Thermo) for 1 hr at room temperature with mixing. The mRNA tag on the fused peptides was reverse transcribed using SuperScript II (Thermo) and a primer that contained (1) a sequence that hybridizes just downstream of variable protein sequence coding region, (2) a 12-nt random sequence that serves as a unique molecular identifier (UMI), and (3) the Illumina read two sequencing primer sequence. Enzyme was heat inactivated by adding EDTA to 10 mM and heating to 65°C for 10 min. To remove aggregates, the library was fractionated in 20 mM Tris pH 8.0, 100 mM KCl, 1 mM MgCl$_2$, 1 mM DTT over a 6 mL 10–30% sucrose gradient by spinning at 60,000 rpm in a SW60 rotor for 10 hr. Fractions of 100–200 µL were collected and the abundance of peptide-mRNA-cDNA fusion in each fraction was assayed using qPCR. The 3–4 peak fractions encompassing 80–90% of total input was pooled and used in pull-down experiments or flash frozen in aliquots and stored at −80°C.

### Pull-down assays

Biotinylated bait proteins (12 µg of Med15 or 8 µg of TFIID subcomplex) were bound to 50 µL Dynabeads MyOne Streptavidin T1 (Thermo) for 15 min at room temperature in binding buffer (20 mM Tris pH 8.0, 100 mM KCl, 10 mM MgCl$_2$, 0.01% Tween-20, 1 mM DTT) and then washed three times for 5 min in binding buffer. In the third wash, beads were blocked by also including 100 ng/µL salmon sperm DNA (Thermo) and 10 ng/µL BSA (NEB) in the buffer. mRNA display library input was prepared by mixing sub-library A, sub-library B, and the three positive and three negative controls. This input was then incubated with the bait-coated beads in binding buffer with salmon sperm DNA and BSA at 100 µL total volume for 8 min with mixing. Afterward, the beads were separated on a magnet, briefly washed once with cold binding buffer, and then immediately separated on a magnet again. cDNA from bound fragments was eluted by suspending the beads in 10 mM Tris pH 8.0 with proteinase K (NEB) and incubated for 30 min at room temperature. Proteinase K was heat inactivated at 95°C for 10 min, beads were separated on a magnet, and the eluate was collected.

Total abundance of the library cDNA in the input, bound, and eluted samples was measured by qPCR. To measure the total abundance of the whole library (without regard to the identity of the library elements), qPCR primers were designed to hybridize to the constant sequences flanking the variable region of the library. Additionally, this design enabled the use of different primers to separately track sub-library A and sub-library B, as well as each of the six control sequences individually. The controls were used to provide an estimate of binding efficiency. Binding percentage was

calculated by dividing the abundance of each library in the bound sample by the abundance in the input sample.

cDNA of the input and bound samples were prepared for sequencing in the same manner as the purified plasmids in the activation assay.

## Surface plasmon resonance (SPR) experiments

Biotinylated 104 bp double-stranded DNA templates were assembled by annealing a 104-nt oligo to a 20-nt oligo with a 5′ biotin modification (IDT) and elongating using Klenow Fragment (3′–>5′ exo-) (NEB). The oligos were designed such that the TF DNA binding sites were placed away from the biotinylated end. SPR experiments were performed on the ProteOn XPR36 Surface Plasmon Resonance System using NeutrAvidin-coated NLC Sensor Chips (Bio Rad). Mass of biomolecules bound to the chip surface was measured in arbitrary units (AU) approximately every 1 s. All plotted data were normalized to the first ligand channel and the first analyte channel which were used as negative controls in experiments as described below.

In SPR experiments with GCN4, 104 bp DNA templates with 0, 1, 2, 4, or 6 Gcn4 motifs were bound to the chip along the ligand channels, except for the first ligand channel, which was left without DNA for normalization. Binding buffer for all conditions was 20 mM Tris-HCl (pH 7.5), 100 mM KCl, 10 mM MgCl$_2$, 2 mM EDTA, 10 mM AmSO$_4$, 0.05% Tween-20, with 1 mg/mL BSA (NEB). Bound DNA was washed twice with 1 M NaCl. Gcn4 at 7.5 nM was then pre-bound to DNA for 480 s along the analyte channels, except for the first analyte channel, which was left without Gcn4 for normalization. After Gcn4 binding, Gcn4 at 7.5 nM with Mediator at 0, 1.25, 2.5, 5, or 10 nM were flowed across analyte channels 2–6 respectively for 300 s, then dissociation was observed for 20 min by flowing in binding buffer. Association, dissociation, and affinity constants of Mediator binding to Gcn4 on DNA were estimated by fitting to a Langmuir kinetic model using the ProteOn Manager Software (Bio Rad). Reported values are the mean and standard deviation across the four different Mediator concentrations. Note that Mediator dissociation rates in conditions with multiple Gcn4 binding sites may be overestimated because it also includes the dissociation of Gcn4 from DNA.

For SPR experiments with EcoRI, biotinylated 60 bp DNA templates containing 0 or 1 EcoRI motif were annealed and elongated as described above, and then bound along the ligand channels and washed with 1 M NaCl. Gcn4AD-EcoRI(E111Q) was bound along analyte channels at 27 nM monomer in 20 mM Tris-HCl (pH 7.5), 10 mM MgCl$_2$, 300 mM NaCl, 10 mM bME, 0.05% Tween-20, 1 mg/mL BSA (higher salt was used to reduce non-specific binding to DNA). Minimal dissociation was observed after binding. Then, Mediator at 5 nM with competitor Gcn4 protein (in molar excess as specified in *Figure 7D*) was flowed across the chip for 180 s in 20 mM Tris-HCl (pH 7.5), 10 mM MgCl$_2$, 100 mM NaCl, 10 mM AmSO$_4$, 10 mM bME, 0.05% Tween-20, 1 mg/mL BSA.

## Protein library design

All protein positions reported in the text are 1-indexed and inclusive. The first library of protein sequences was composed of sub-library A, which contained wild-type tiles spanning all *S. cerevisiae* transcription factors, and sub-library B, which contains mutants of 8 known *S. cerevisiae* ADs. The second library was composed of sub-library D and sub-library E, which contained multiple collections of wild-type, mutant, and designed sequences. Sequences in sub-libraries A, B, and D were 53 amino acids in length and sequences in sub-library E were 30 amino acids in length. Protein and DNA sequences, as well as activation and pull-down data where available, are listed in Tables S1 and S4. Libraries contained additional fragments which were included in experiments but are not reported here.

### Sub-library A

Sub-library A had 7637 fragments in total. This peptide library was generated from all 162 yeast (*Saccharomyces cerevisiae* S288C) transcription factors—that is genes annotated with the Gene Ontology term GO:003700 (DNA-binding transcription factor activity) (*Ashburner et al., 2000*; *The Gene Ontology Consortium, 2019*)—and also MET4 and HAP4. Each transcription factor protein sequence was fragmented into 53-amino acid tiles with at least 40 amino acid overlap (overlap was adjusted based on the protein length to make the tiling as evenly distributed as possible across any given protein), yielding 7460 unique peptides. A set of 50 random synthetic (i.e. not derived

from a known protein) sequences with the same amino acid frequencies as the overall set of yeast transcription factors, as well as a set of 50 peptides derived from non-nuclear proteins (proteins not annotated with the GO term GO:0005634 [nucleus]) were used as negative controls. Additionally, 10 positive control sequences from known ADs, some of which overlapped TF tiles, were included. These control protein sequences were each synonymously encoded by eight different DNA sequences in this sub-library (see below); the mean activation across the eight encodings was used in further analysis. Data involving this sub-library is shown in *Figure 1C–E* and *Figure 1—figure supplement 1C–D and G–J*; *Figure 2A* and *Figure 2—figure supplement 1D*; *Figure 3B* and *Figure 3—figure supplement 1A–C and J–K*; *Figure 5B–D* and *Figure 5—figure supplement 1B–K*.

## Sub-library B

Sub-library B had 2900 fragments in total. Eight 53-residue regions from previously reported ADs were selected for more detailed analysis with mutagenesis. These regions were from Gcn4 (82-134), Gal4 (829-end), Met4 (72-124), War1 (892-end), Hap4 (424-476), Pho4 (58-110), Pdr1 (1016-end), and Rap1 (630-682). The wild-type sequence of the eight regions were synonymously encoded by eight different DNA sequences. For each region, similar classes of mutants were designed, as follows.

- Scanning mutants: Each position was individually mutated either to alanine or glycine. Used in *Figure 2—figure supplement 1F–G*.
- Scramble mutants: Eight scrambles of each region were randomly generated. Used in *Figure 3A* and *Figure 3—figure supplement 1A–B*.
- Aromatic mutants: Up to 10 different subsets of $N$ aromatic residues (Trp, Phe, or Tyr) were randomly selected and mutated to alanine, where $N$ ranged from one to the total number of aromatic residues. Used in *Figure 2D*, *Figure 5E*, *Figure 2—figure supplement 1H*, and *Figure 3—figure supplement 1A–B*. Also, 1–4 non-hydrophobic positions were randomly selected to mutate to Trp, Phe, or Leu, in five ways each.
- Acidic mutants: Up to three different subsets of $N$ acidic residues (Asp or Glu) were randomly selected and mutated to alanine, where $N$ ranged from one to the total number of acidic residues. Used in *Figure 2D*, *Figure 5E*, *Figure 2—figure supplement 1H*, and *Figure 3—figure supplement 1A–B*. Also, 1–10 non-hydrophobic positions were randomly selected to mutate to Asp, in three ways each.
- Homologs: Proteins from other fungal species homologous to *S. cerevisiae* (yeast) TFs were found by protein alignment to the NCBI nr (non-redundant protein sequence) database using BLAST. A multiple sequence alignment (MSA) was then constructed for each yeast TF and its homologous fungal proteins using Clustal Omega. The homologous fungal proteins were then filtered for those that had at least one match in the MSA to a known yeast ADs (i.e., the region aligned to the known AD was not entirely spanned by a gap). This yielded 285, 251, and 179 homologous sequences for the Gcn4 (120-134), Pho4 (81-110), and Pdr1 (1044–1068) ADs, respectively. Used in *Figure 3—figure supplement 1A–B*.

Two additional sets of 50 random synthetic sequences and 50 peptides derived from non-nuclear proteins were all also included as negative controls, as described for sub-library A.

## Sub-library D

Sub-library D had 2972 unique fragments in total. To assess reproducibility, this sub-library included the strongest activating fragment from each of the 150 ADs identified and the 50 random sequence controls from sub-library A, which are shown in *Figure 1—figure supplement 1G*.

- AD mutants: For each fragment from the 150 ADs, mutants were included that (1) mutated all Asp to Asn and all Glu to Gln, (2) mutated all Asp to Glu, or (3) mutated all Glu to Asp. Used in *Figure 2B* and *Figure 2—figure supplement 1C and E*.
- Tandem cADs: Analysis of sub-library A data enabled identification of cADs. Forty-seven pairs of identified cADs that could fit within a 53-aa fragment in their native context were chosen. Fragments for this sub-library were designed such that they each contained one of these pairs of cADs, with their original distance between them preserved. Additionally, the first, second, or both cADs were embedded in neutral sequence consisting of AGSTNQV residues, in a manner that maintained the original distance between the two cADs. Used in *Figure 7A*.
- Hydrophobic inhibition matrix: A 20-aa variable region was placed five residues upstream of the 13-aa Pdr1 cAD and embedded within neutral sequence to reach 53 residues total.

Random sequences for the variable region were generated, such that relative amino acid frequencies were representative of those from yeast TFs, except for FEKR and WFYLM residues, whose frequencies were systematically inflated to generate a wider range of net charge and hydrophobicity. Eight fragments fitting into each bin were randomly selected to be included in the library. Used in *Figure 4J*.

- ADs in yeast nuclear proteins: Regions of non-TF nuclear proteins that were predicted by PADDLE to activate were included. Additionally, highly hydrophobic and acidic regions that were predicted by PADDLE not to activate were also included. All designs involving PADDLE predictions are described in more detail below. Used in *Figure 7F* and *Figure 7—figure supplement 1G*.

## Sub-library E

Sub-library E had 3621 unique fragments in total. Fragments in this sub-library were all 30 residues in length. Designed sequences of interest, either 13 aa or nine aa in length, were located at the center of these fragments. The regions flanking these sequences were constant: ASAVGQQG[AN] (upstream) and [NS]SATQSQSNT (downstream), with brackets indicating additional residues for the 9-aa designed sequences.

- AD tiling: Ten identified ADs were selected (from analysis of sub-library A data), and 13-aa tiles spanning each one at 1-aa spacing were included in this sub-library. Used in *Figure 3E* and *Figure 2—figure supplement 1F*.
- Core AD scrambles: The 28 cADs 13 aa in length with the strongest predicted activation (identified from analysis of sub-library A data) were included. Additionally, 33 different scrambled sequences of each cAD were included. Used in *Figure 4A–B and D*, and *Figure 4—figure supplement 1A*.
- Proline mutants: In each of the 28 selected cADs, non-WFYILM residues in positions 4–10 were individually mutated to either Ala or Pro. Used in *Figure 4C–D*, *Figure 4—figure supplement 1B*, and *Figure 6—figure supplement 1B*.
- Systematic 9mers: All sequences nine residues in length consisting of only two amino acids each—Asp and either Leu, Phe, or Trp—were included, totaling $3 \times 2^9 - 2 = 1534$ sequences. Used in *Figure 4E–H* and *Figure 4—figure supplement 1C–E*.

Fifty random sequence negative controls were included; these were the first 30 residues of the controls from sub-library B.

## Design of DNA sequences encoding proteins

In the reverse translation design process, we aimed to optimize our library DNA fragments for compatibility and consistency with our in vitro and in vivo assays, as well as with standard RNA-seq protocols. We also sought to build in redundancy for error-correcting reads. In particular, we used the Python package dnachisel 1.4.1 (*Zulkower and Rosser, 2020*) to optimize the following objectives:

1. Use codons matching the relative frequencies in the rabbit species corresponding to the in vitro translation kit. Codon frequencies were pulled from the Codon Usage Database hosted by the Kazusa DNA Research Institute (*Nakamura et al., 2000*).
2. Target an optimal GC content of 45% at both a local (sliding window of 50) level and a global (entire fragment) level.
3. Avoid repeated subsequences of length 10 or more.
4. Avoid homopolymer runs of 8 or more As, eight or more Ts, five or more Cs, or five or more Gs.
5. Avoid adjacently repeated k-mers, specifically 3-peats of 3-mers or 5-peats of 2-mers.

To allow paired-end sequencing to uniquely identify each element, we additionally enforced an edit distance of 6 among the first 48 bases and last 48 bases of any two sequences in the same sub-library. This was performed in a randomized, brute-force, iterative approach, with each iteration consisting of the following steps:

1. Pairwise edit distances were computed for all sequences in the sub-library.
2. For any sequence with edit distance less than six from another sequence, two codons in the first 48 bases and two codons in the last 48 bases were randomly selected and changed while respecting the encoded amino acid sequence.

3. Repeat until no changes are needed.

Finally, we verified that all sequences sharing the same sub-library primer (e.g. all sequences in sub-library A, or all sequences in sub-library B) had a paired-end edit distance (sum of edit distance of 5'-most 50 bases and edit distance of 3'-most 50 bases) of at least 6.

## Sequencing read alignment

The use of sequencing primers unique to each sub-library enabled us to submit samples for sequencing in multiplexed format and accurately assign reads to the correct sub-library computationally. We further leveraged the edit distance margin built into the library to enable mapping of sequencing reads with a small number of errors.

Sequencing read alignment was performed using custom bash script built on top of existing tools and additional custom scripts. It takes as input arguments the UMI length, the sub-library sequencing primer, the edit distance threshold for that sub-library, and raw FASTQ files. For pull-down experiments, unique molecular identifiers (UMI) were extracted from reads and appended to the read names using umi_tools 1.0.0 (*Smith et al., 2017*). cutadapt 1.18 was used to discard reads without matching paired-end sub-library sequencing primers and trim the primers in reads with matching primers; the default error tolerance was used (*Martin, 2011*). bwa-mem 0.7.17-r1188 was used to perform a first-pass alignment of reads to the DNA fragment library (*Li, 2013*). Imperfectly mapped read pairs (i.e. those without paired read SAM flags of 99 and 147) were re-mapped to the library sequence with minimal edit distance. This was necessary because bwa-mem did not always correctly map paired reads as a pair, a problem most evident in the mutant library with many similar sequences. Pairwise Levenshtein edit distance was computed using the Python package editdistance 0.5.3 (https://github.com/roy-ht/editdistance, *Tanaka, 2019*). Paired reads exceeding the edit distance threshold were discarded using reformat.sh from BBTools 38.61 (https://sourceforge.net/projects/bbmap/). Duplicate reads were identified and deduplicated using utmi_tools 1.0.0. Finally, reads mapped to each DNA library fragment sequence were counted.

## Activation assay data processing

Calculations of GFP signal and fold-activation required the following inputs:

1. The total number of cells sorted into each bin
2. The total number events assayed when sorting bins 1–4 versus bins 5–8
3. The number of sequencing reads of each GFP bin mapped to each fragment
4. The total number of sequencing reads obtained for each bin
5. The average GFP signal observed in each bin during sorting
6. The average GFP signal and number of cells observed for each bin for a negative control population.

First, to make cell counts comparable between sorting bins 1–4 versus bins 5–8, the number of cells sorted into each GFP bin were normalized based on the number of events assayed in each sort. Second, to estimate the number of cells sorted into each of the 8 GFP bins that expressed a given protein fragment F, the total number of cells sorted into each bin was multiplied by the fraction of sequencing reads for that bin that mapped to F. Any fragment represented by fewer than 10 cells were excluded from further analysis. Third, the average GFP signal of cells expressing F was calculated as a weighted average from the average GFP signal observed in each bin during sorting weighted by the distribution of the number of cells expressing F across the eight bins. Because GFP distributions cells expressing single AD sequences were normally distributed in FACS data when GFP signal (arbitrary units) was plotted in log-scale, all averages of GFP distributions were calculated in log-scale (equivalently, a geometric mean of GFP values). The standard deviation of GFP signal of cells expressing F was similarly calculated in log-scale and used as a metric for measurement precision. All averages involving activation reported in the paper are calculated in log-scale (equivalently, a geometric mean). Fourth, the fold-activation of F was calculated as the mean GFP signal of F divided by the mean GFP signal of negative control cells (see below). Z-scores were calculated by standardizing the GFP signal of negative control cells to mean 0 and variance one in log scale and p-values were calculated with one tailed Z-tests.

The mean and standard deviation GFP signal of negative control cells was defined in one of two ways. In the experiment with sub-libraries A and B, a sample of estrogen-induced cells containing

only the empty pRS414-TFchassis plasmid was assayed under identical conditions and the mean and standard deviation GFP signal was computed from FACS data. In the experiment with sub-libraries D and E, these values were instead determined internally from the 50 random sequence negative controls in the pooled library. Since a small fraction of random sequences activate, outlier sequences were iteratively excluded if they had GFP signal greater than three standard deviations away from the mean, resulting in three sequences excluded from each sub-library. Then mean and standard deviation were computed from the sum of the negative control GFP distributions.

In activation assays with sub-libraries D and E, we noticed that many sequencing reads that were mapped with a small number of errors corresponded to fragments with a very large GFP standard deviation. Upon closer examination of sequencing reads of example fragments, the same sequencing errors were consistently found, which likely corresponded to mutations that occurred in vivo that affected function of the protein fragment (e.g. inactivation of a strong AD by a frameshift mutation). We therefore used only reads that aligned perfectly for this experiment. Furthermore, fragments with GFP standard deviation greater than 4.5 were excluded from further analysis.

The following three metrics were tracked: the number of total unique fragments, number of fragments observed in at least one of the eight GFP bins, and number of fragments that passed filtering. For the activation assay with sub-libraries A and B, this was (7637, 7604, 7549) for sub-library A and (2900, 2833, 2813) for sub-library B. The same metrics for the activation assay with sub-libraries D and E was (7733, 7716, 7345) for sub-library D and (4261, 4261, 4252) for sub-library E. These numbers include additional fragments in the peptide libraries that are not reported in this paper.

## Activation domain annotations

To define ADs across all TFs, at each protein position a mean Z-score was calculated from the Z-scores of all fragments overlapping that position. ADs were defined for all intervals of the protein that had a mean positional Z-score (MPZ) greater than 3.5. Starting with intervals that had the highest maximal MPZ score, the start and stop positions of the AD were defined by the full width half maximum; that is, the positions in the protein at which the MPZ fell below half of the maximal MPZ value in that interval. If the AD defined in this manner includes a new region with an MPZ score higher than the original maximal value, this indicated that the MPZ peak was only small relative to nearby peaks, so the AD was then excluded.

## Activation data analysis

We compared our ADs with the Induction Dynamics gene Expression Atlas (IDEA) (*Hackett et al., 2020*), which induced expression of every TF individually and measured 'v_inter' parameters that describe the resulting fold-change in the expression of all target genes. We said that a TF activated a gene if the v_inter > 1, and for each TF that regulated at least five genes, we calculated the proportion of regulated genes that were activated. We then compared these proportions for TFs that had a strong AD (overlapping a fragment with Z-score at least 6) versus TFs without ADs. Significance was calculated using a Kolmogorov–Smirnov test.

Hydrophobicity measurements were calculated using the Wimley-White interfacial scale, which measures whole-residue $\Delta$G for transfer from water to a water-octanol interface (*Wimley and White, 1996*). Because the goal was to quantify the total content of hydrophobic residues rather than the net hydrophobicity/hydrophilicity, hydrophobic content of a peptide is computed as the sum of the scores of only the hydrophobic residues (i.e. those with negative $\Delta$G). Net charge was computed by adding the number of Arg and Lys residues and subtracting the number of Asp and Glu residues.

Motif finding using DREME was done through the web portal (http://meme-suite.org/tools/dreme) using default settings and shuffled input sequences as control (*Bailey, 2011*). Searching for 9aaTAD sequences was done by matching to the 'most stringent pattern' [MDENQSTYG]{KRHCGP}[ILVFWM]{KRHCGP}{CGP}{KRHCGP}[ILVFWM][ILVFWMAY]{KRHC} listed at the prediction portal https://www.med.muni.cz/9aaTAD/index.php (*Piskacek et al., 2007*).

## Comparison with ADpred

ADpred was installed locally from https://github.com/FredHutch/adpred, *Erijman, 2020a* and predictions were computed for all 30-aa tiles on all yeast TFs (*Erijman et al., 2020b*). ADs predicted by ADpred were defined using the authors' criteria, namely sites with five or more contiguous residues

with a score >= 0.8. An ADpred-predicted region was considered correct if it overlapped by at least five aa with a significantly activating tile (p<0.0001). With this approach, ADpred achieved a precision and recall of 0.366 and 0.692 in predicting experimental ADs, respectively. When ADpred-predicted regions were randomly shuffled 100 times to different proteins and positions, the average precision and recall of these shuffled predictions were 0.120 and 0.295 respectively. Alternatively, to compare ADpred predictions directly to our experimental measurements of activation, a representative ADpred score for each 53-aa TF tile in our library was calculated by taking the 80th percentile of ADpred scores of 30-aa tiles contained within the 53-aa tile. These scores were used for calculating tile-wise precision-recall curves for ADpred predictions (*Figure 3—figure supplement 1K*).

## mRNA display pull-down data processing

The fractional pull-down of each fragment $F$ was calculated as (the fraction of the total input library that bound to beads, measured from qPCR) x (the fraction of UMI-deduplicated sequencing reads in the bound sample matching $F$) / (the fraction of UMI-deduplicated sequencing reads in the input sample matching $F$). Fragments with fewer than 30 reads in the input sample or fewer than five reads in the bound sample were excluded from downstream analysis. Baseline binding was defined using the 50 random protein control sequences in each library, except the sequences that were outliers in the activation assay were excluded. Specifically, random control outliers were iteratively removed if their activation was more than three standard deviations away from the mean (in log scale), resulting in the exclusion of random control #2, 12, 16, 38 from sub-library A and #17, 18, 21, 22, 31, 43 from sub-library B. Binding enrichment of each fragment was then computed as the fractional pull-down of the fragment divided by the mean fractional pull-down (in log scale) of the random controls. Because fractional pull-down and enrichment were normally distributed in log-scale, all averages involving pull-down or enrichment reported in the paper are calculated in log-scale (equivalently, a geometric mean). Z-scores were computed by standardizing the binding enrichment of random controls to mean 0 and variance one in log-scale and P-values were calculated with one tailed Z-tests. Triplicate measurements for Med15 pull-downs and duplicate measurements for TFIID subcomplex pull-downs were combined by averaging the binding enrichment values. A small number of highly positively-charged fragments bound in Med15 pull-downs when the mRNA display library was purified on a sucrose gradient but did not bind when the sucrose gradient was omitted. Suspecting that this binding was artifactual, potentially through non-specific binding to mRNA tags, we excluded from analysis 92 fragments that bound with Z-score between −2.5 and 1 without the sucrose gradient and had a Z-score at least 3.5 higher with the sucrose gradient than without.

## Overlap of ADs and Med15- and TFIID subcomplex-binding domains

Med15-binding domains were annotated in the same manner as ADs using a mean positional Z-score, but with a cutoff of 2.23. An AD was considered to bind Med15 or TFIID subcomplex if it overlapped a fragment that bound significantly (p<0.001) by at least 26 aa. A Med15-binding domain was considered to activate if it overlapped an AD.

## Neural network training

Neural network models were trained to predict activation Z-scores. Approximately 15% of sequences from sub-library A and B were randomly held-out as a test set, and the remaining library elements were split into 10 parts for training and cross-validation. Because adjacent TF tiles have significant overlap in sequence, all tiles from each protein were placed into the same train or test split. From sub-library B, all mutants and homologs of the Pdr1 AD and all scramble mutants were held-out in the test set and the remaining sequences were distributed evenly across the training splits. Models were developed in TensorFlow 2.2 (*Abadi et al., 2016*) and trained using CPUs on Stanford's Sherlock computing cluster. All models used mean squared error as the loss function. Sequence encodings and activation Z-scores were standardized to mean 0 variance one across the training dataset for more efficient learning. For each architecture, 10 models were trained on each subset of nine training splits and validated on the 10th split. The best model architecture was chosen by the mean validation score across the 10 splits on sub-library A sequences only. For final evaluation on the test dataset, the mean prediction across all 10 models was used.

## Predicting activation from amino acid composition

Protein fragment sequences were each encoded as a single 20-element vector giving the proportions of each of the 20 amino acids. The best architecture, as determined by a grid search over hyperparameters, was a neural with three fully connected layers of width 40 with Swish activation (*Ramachandran et al., 2017*), batch normalization, L2 weight penalty of 0.01, and dropout rate of 0.4. The model was trained with an initial learning rate of 0.0001 using the Adam optimizer for up to 300 epochs with two scheduling callbacks: reduction of the learning rate by fivefold if training loss did not improve for 20 epochs, and early stopping if no improvement on the validation loss (averaged in a 10-epoch window) was observed for 75 epochs, upon which the best model from previous epochs was saved.

## Predicting activation from protein sequence (PADDLE)

One-hot encodings were used to encode each 53-aa protein fragment sequence as a 53-by-20 matrix. Secondary structure predictions were generated using PSIPRED v. 4.01 without PSI-BLAST (*Jones, 1999*), and a 53-by-2 matrix of the alpha helix and coil predictions were also used as input. Disorder predictions were generated using IUPred2A (*Mészáros et al., 2018*), and a 53-by-2 matrix of the disorder predictions in 'short' and 'long' mode were also used as input. In total, each 53-aa protein fragment was encoded as a 53-by-24 matrix.

The best model architecture, as determined by a grid search over hyperparameters, was a convolutional neural network with nine convolutional layers with kernel size 10 and channel width 30 (i.e. number of filters) followed by max-pooling along the sequence-length dimension and two fully-connected layers of width 20, with Swish activation. For regularization, the L2 weight penalty was 0.001, dropout rate was 0.1, and batch normalization was used. The model was trained with an initial learning rate of 0.001 in the same manner as the amino acid composition-based network, except for up to 500 epochs.

After finding this optimal architecture, the test dataset was distributed across the 10 training datasets and 10 new models were trained, together comprising PADDLE. The mean value of these 10 predictors was used as the final prediction. This model was used for all predictions on yeast TFs and nuclear proteins, human TFs, and virus transcriptional regulators. Because secondary structure prediction was by far the slowest aspect of running PADDLE predictions, we trained a version of PADDLE called PADDLE-noSS which only used the one-hot sequence encoding for input, no secondary structure or disorder predictions. PADDLE-noSS was nearly as accurate as PADDLE ($R^2$ scores of 0.76, 0.52, and 0.89 for TF tiles, scramble mutants, and Pdr1 mutants and homologs respectively; compare to *Figure 3—figure supplement 1B*) and was used for predictions of core ADs and mutant sequences in *Figure 3E–G*, *Figure 3—figure supplement 1E–I*, and *Figure 4—figure supplement 1F*.

## PADDLE predictions

Human TFs were defined by Gene Ontology term GO:003700 (DNA-binding transcription factor activity). Virus transcriptional regulators (vTRs) were taken from *Figure 1—source data 1* from *Liu et al., 2020*. Within each human TF or vTR, predictions on 53-aa tiles at 1-aa resolution were generated using PADDLE and smoothed with a 9-aa moving average. High-strength ADs were defined as the union of at least 5 consecutive 53-aa tiles with a predicted Z-score greater than 6. ADs defined in this way that overlapped by more than 26 aa were combined. Medium-strength ADs were defined similarly with a Z-score cutoff of 4 and were required to not overlap with high-strength ADs by more than 26 aa. All protein positions reported in the text are 1-indexed and inclusive; all annotations reported in supplemental tables are 0-indexed and inclusive of the start position but exclusive of the stop position. For testing in a luciferase assay, 50 predicted high-strength ADs from human TFs were chosen at random and the tile with the strongest (smoothed) predicted Z-score from each AD was used.

Yeast nuclear proteins were defined by Gene Ontology term GO:0005634 (Nucleus). Within each protein, predictions on 53-aa tiles at 1-aa resolution were generated using PADDLE and smoothed with a 9-aa moving average. All predicted Z-score local maxima higher than 3.09 (p<0.001) were considered for experimental testing. The 53-aa tiles corresponding to the local maxima were included, starting from those with the largest predicted Z-score, unless they overlapped a

previously-included tile by at least 15 aa. Analysis of experimentally-confirmed ADs focused on proteins that had experimental or high-throughput evidence for nuclear localization. Proteins involved in transcriptional regulation were defined by GO:0006355 (regulation of transcription, DNA-templated). To calculate enrichment of activating tiles in coactivator proteins, we generated a null model in which 'activating' tiles, equal in number to actual activating tiles discovered in our screen, were randomly selected from among non-TF nuclear proteins. We then counted how many of these randomly-selected tiles were in coactivator proteins, and then repeated this sampling 10,000 times to generate a null distribution. The actual number of tiles, 42, is 2.9-fold more than the mean of the null; $p < 10^{-12}$ by one tailed Z-test. Predicted disorder for each tile was computed from D2P2 annotations as the fraction of the tile's residues that were annotated as disordered among all 8 of the predictors listed in D2P2.

To define core ADs, predictions of sequences shorter than 53 aa were done using PADDLE-noSS by embedding the shorter region in 100 different neutral sequences composed of AGSTNQV residues and averaging the predicted values. Predictions of all tiles 5–30 aa in length within every AD were computed in this manner, and used to define core ADs for *Figure 3F–G*, *Figure 7A*, and *Figure 7—figure supplement 1B*. Specifically, core ADs were defined as the shortest tiles with predicted Z-score greater than 3.09 (p<0.001) that do not overlap another shorter core AD. In *Figure 3—figure supplement 1I*, the 20-aa region within each AD with the highest predicted Z-score was included if it had Z-score greater than 3.09, and the effects of single AA mutants in these 20-aa core ADs were predicted in the same manner using PADDLE-noSS.

Code for running PADDLE and PADDLE-noSS is available at https://github.com/asanborn/PADDLE (*Sanborn, 2021*; copy archived at swh:1:rev:17aa36985e474d8593b99d593f7cc5e7-c3e205a0). Pre-generated PADDLE predictions from multiple species are available for download at http://paddle.stanford.edu.

## Computational peptide docking using FlexPepDock

To generate structural models of binding, we took 28 core ADs peptides 13 residues in length and docked them to the ABDs using Rosetta3 (*Leaver-Fay et al., 2011*), specifically its FlexPepDock ab initio pipeline (*Raveh et al., 2011*). Core ADs were used as the peptide, and activator-binding domain as the receptor. Peptides were initialized in an extended conformation using Rosetta's BuildPeptide application and manually placed near the binding pocket. For the KIX domain, NMR chemical shift data was used to aid in placement (*Thakur et al., 2008*). For ABD1, the AD of Gcn4 was used to aid in placement. For all models of ABD1, the first conformer from PDB structure 2LPB was used (*Brzovic et al., 2011*).

For each AD-ABD pair, a set of 50,000 structural models were generated and ranked by FlexPepDock's reweighted_sc score. Binding poses were generated by clustering the top 500 ranked models using Rosetta's cluster application with the default clustering distance threshold of 2 Å. The best-scoring model from each cluster was designated as the cluster representative, and the cluster representatives that were also among the 10 best-scoring models were used for further analysis. Additionally, larger runs with 500,000 structural models and clustering on the top 5000 models were run for six individual ADs for the KIX ABD. The full list of ABDs and ADs used is available in *Figure 6—source data 1*. PDB files of 10 best-scoring models from different binding poses and the score, cluster number, and RMSD to best model for the 500 best-scoring models are available in *Figure 6—source data 2*. Due to the large number of peptides that needed to be docked, the FlexPepDock *ab-initio* pipeline was modified to run in a cluster setting at a large scale and used approximately 100,000 CPU hours on the Stanford Sherlock academic cluster.

Secondary structure was calculated using DSSP (*Kabsch and Sander, 1983*; *Touw et al., 2015*) and annotations of G, H, or I were considered alpha-helical. Interaction types were computed with GetContacts (*Venkatakrishnan et al., 2019*) and per-AA accessible surface areas were computed with FreeSASA (*Mitternacht, 2016*). For each final docked structural model, buried peptide surface area was taken by subtracting the accessible surface area of the peptide in complex with the receptor from the accessible surface area of the peptide on its own. For *Figure 6E*, marked positions correspond to atom CG of Asp, atom CG of Leu, and the midpoint of the CG and CZ atoms in Phe (the center of the aromatic ring). Residues are shown from all the 10 best models from different clusters for all core AD structures if the marked position is within 3 Angstroms of the KIX domain or ABD1 structure.

Adapted FlexPepDock code, and wrapper code for GetContacts and FreeSASA are available at https://github.com/drorlab/med15.

## Acknowledgements

We thank Ramon Lorenzo Labitigan, Peter Geiduschek, and members of the Kornberg lab for critical feedback and discussions; Ramon Lorenzo Labitigan for assistance with the manuscript and figures; Ralph Davis and Shigeki Nagai for providing purified Mediator; Barbara Davis for assistance with experiments; Namita Mitra and Zane Colaric for sequencing; Ricardo Zermeno and the Stanford Shared FACS Facility for cell sorting; the Weis lab for sharing tissue culture facilities; Rich Roberts and Terry Takahashi for advice on mRNA display; the Stanford Research Computing Center for providing computational resources and support; and Shantao Li, Haiwen Gui, Sarah Gurev, Avanti Shrikumar, and Christopher Yeh for help with machine learning. Stanford may seek to commercialize aspects of this work, and related applications for intellectual property have been filed.

## Additional information

### Funding

| Funder | Grant reference number | Author |
|---|---|---|
| National Institutes of Health | R01-DK121366 | Roger D Kornberg |
| U.S. Department of Energy | Scientific Discovery through Advanced Computing (SciDAC) program | Ron O Dror |
| National Institutes of Health | 4D Nucleome Grant (U01HL130010) | Erez Lieberman Aiden |
| National Institutes of Health | Encyclopedia of DNA Elements Mapping Center Award (UM1HG009375) | Erez Lieberman Aiden |
| National Science Foundation | PHY-2019745, Center for Theoretical Biological Physics | Erez Lieberman Aiden |
| Welch Foundation | Q-1866 | Erez Lieberman Aiden |
| McNair Medical Institute | Scholar Award | Erez Lieberman Aiden |
| United States - Israel Binational Science Foundation | 2019276 | Erez Lieberman Aiden |
| IBM Corporation | In-kind support | Erez Lieberman Aiden |
| U.S. Department of Defense | National Defense Science & Engineering Graduate (NDSEG) Fellowship | Adrian L Sanborn |
| U.S. Department of Energy | Office of Science Graduate Student Research (SCGSR) program (DE-SC0014664) | Raphael JL Townshend |
| National Institutes of Health | F32-GM126704 | Jordan T Feigerle |
| Illumina | In-kind support | Erez Lieberman Aiden |
| National Institutes of Health | R01-AI021144 | Roger D Kornberg |
| National Institutes of Health | 4D Nucleome Grant (U01HL156059) | Erez Lieberman Aiden |
| National Science Foundation | PHY-1427654, Center for Theoretical Biological Physics | Erez Lieberman Aiden |
| National Science Foundation | DBI-2021795, Behavioral Plasticity Research Institute | Erez Lieberman Aiden |

The funders had no role in study design, data collection and interpretation, or the decision to submit the work for publication.

## Author contributions
Adrian L Sanborn, Conceptualization, Data curation, Software, Formal analysis, Supervision, Investigation, Visualization, Methodology, Writing - original draft, Writing - review and editing; Benjamin T Yeh, Data curation, Software, Formal analysis, Methodology; Jordan T Feigerle, Investigation; Cynthia V Hao, Data curation, Investigation, Visualization; Raphael JL Townshend, Data curation, Software; Erez Lieberman Aiden, Resources; Ron O Dror, Resources, Supervision; Roger D Kornberg, Conceptualization, Resources, Supervision, Funding acquisition, Writing - review and editing

## Author ORCIDs
Adrian L Sanborn (iD) https://orcid.org/0000-0002-4725-8012
Benjamin T Yeh (iD) http://orcid.org/0000-0001-9397-6392
Cynthia V Hao (iD) http://orcid.org/0000-0003-2183-0698
Roger D Kornberg (iD) https://orcid.org/0000-0002-2425-7519

## Decision letter and Author response
Decision letter https://doi.org/10.7554/eLife.68068.sa1
Author response https://doi.org/10.7554/eLife.68068.sa2

## Additional files

### Supplementary files
• Transparent reporting form

### Data availability
All data from in vivo activation and in vitro screens are included in tables as source data files. PDB files of structural models of Med15-AD interactions are included in Figure 6-source data 2. All sequencing data have been deposited in GEO, under the accession code GSE173156.

The following dataset was generated:

| Author(s) | Year | Dataset title | Dataset URL | Database and Identifier |
|---|---|---|---|---|
| Sanborn AL, Yeh BT, Feigerle JT, Hao CV, Townshend RJL, Aiden EL, Dror RO, Kornberg RD | 2021 | Simple biochemical features underlie transcriptional activation domain diversity and dynamic, fuzzy binding to Mediator | https://www.ncbi.nlm.nih.gov/geo/query/acc.cgi?acc=GSE173156 | NCBI Gene Expression Omnibus, GSE173156 |

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
