## [Decision Letter]

**Acceptance summary:**

This study is remarkable for its breadth of analysis to elucidate the important sequence composition and structural features of yeast transcriptional activation domains and the nature of their biochemical interactions with the Med15 subunit of the coactivator Mediator. It also presents biochemical findings supporting the notion that interactions of multiple activation domains of a transcription factor with a multiplicity of activator binding domains within Mediator cooperate to ensure stable recruitment of Mediator and efficient stimulation of transcription initiation.

**Decision letter after peer review:**

Thank you for submitting your article "Simple biochemical features underlie transcriptional activation domain diversity and dynamic, fuzzy binding to Mediator" for consideration by *eLife*. Your article has been reviewed by 3 peer reviewers, one of whom is a member of our Board of Reviewing Editors, and the evaluation has been overseen by Kevin Struhl as the Senior Editor. The reviewers have opted to remain anonymous.

Essential revisions:

Revise the manuscript text or figures to address the following comments:

1. Text should be revised to eliminate overreaching statements claiming that all ADs or core ADs have been identified here, and that the principles governing the function of all ADs have been identified. Text and figures should also be modified to make it clear when conclusions are based on PADDLE predictions versus experimentally analyzed mutations of ADs. Several overreaching statements should be qualified regarding conclusions that recruitment of Mediator through Med15 is the predominant interaction between ADs and cofactors driving transcriptional activation in vivo, and that recruitment of TFIID is secondary at best. The relevant locations in text are as follows:

– line 64 and 120: Given that the tiling approach in Figure 1 identified only 75-85% of previously reported ADs, these are likely to be overstatements.

– Figures 3F-3G, and Figure S7B should be labeled to make it clear that the ADs being analyzed were predicted; and the text in lines 215-219 should be revised to make it clear that the mutations were generated in silico and their effects on activation predicted by PADDLE.

– lines 218-219 and line 222: This is an overstatement as many of the conclusions are based on predictions by PADDLE, and might not stand up to experimental testing. The words "all" and "every" should be replaced by "many".

– 233-235: this is an overstatement as only 28 13-amino acid cADs were analyzed and only 19 of these appear to activate by the composition-driven mechanism.

– lines 250-252: Should be qualified to say that these features generally hold, as there are some grey areas in the results of Figure S4B, with some supposedly composition-driven cADs in which proline substitutions had significant effects, eg. War1 and Oaf3.

– lines 346-349: This seems like an overstatement as it is quite possible that other coactivators will contain subunits with ABD segments capable of the fuzzy interactions identified by Hahn and colleagues for the Med15 ABDs.

– lines 361-364: This is an overstatement, as there are certainly some TF tiles that bound TFIID and Med15 with similar affinity in Figure S4I-K. Also, these in vitro pulldown assays might not reflect recruitment in vivo or the extent to which TFIID vs. Mediator provides coactivator function at particular promoters.

2. Line 180-181: Was this true only for one AD, Arg81? Lines 231-232: Indicate the correlation coefficient and associated p-value for the plot in Figure S4A.

3. The definitive manner in which the authors state that they used 'all yeast TFs' (164) as their starting point requires explanation. Where do these come from? Who decided this was the complete list? The list of TFs used should be provided and its origin disclosed.

4. At the end of page 5, line 101-102, the authors allude to cells being assay per tile and 99 being 'seen' in at least 10 cells. This may require better explanation for the uninitiated. At least this reviewer found it difficult to comprehend what was meant.

5. In Figure 1D, please mark the positions of known (previously characterized) ADs.

6. In 1F, a large proportion of the TFs do not contain an AD. This requires a comment or proposed explanation.

7. The references to the Rosetta suite programs on page 26, line 373 is confusing. It should be moved to line 272 (after 'Rosetta Suite').

8. Lines 135-138. The description of the relative abilities of Asp and Glu to promote activation and the results in Figure 2 figure supplement 1D is a little confusing to me. Would it be appropriate to say that Asp is more strongly associated with activation by wild type tiles than Glu, rather than saying that Asp but not Glu is associated…? If not, the authors should comment further on their observation that in a fair number of cases ADs with Asp residues mutated with Glu exhibit reduced but still substantial activation activity.

9. Line 311. The authors should describe Med15 as the primary Mediator subunit targeted by "many ADs," rather than implying that it is the primary subunit targeted by all. As shown in Figure 5C, while a remarkable 73% of ADs tested do bind MED15, there are still a significant number that don't.

10. Lines 386-388 and Figure 6, figure supplement 1B. The authors should comment on the group of cAD domains whose activities are minimally affected (<3-fold) by PRO insertion but predicted by modeling to have a high percentage of residues in alpha helix when bound to KIX domain.

11. Line 991, legend to Figure 3, Supplement 1, panel A. I suspect the authors meant to refer to panel B, rather than C, for comparable PADDLE predictions.

12. Legend to Figure 6A. the legend refers to hydrophobic (yellow) and acidic (green) residues that form the AD-binding surfaces displayed. In the figure green residues are labeled as basic.

13. Consider revising the text to address the Recommendations to authors provided by each of the reviewers.

*Reviewer #1:*

Text should be revised to eliminate overreaching statements claiming that all ADs or core ADs have been identified here, and that the principles governing the function of all ADs have been identified. Text and figures should also be modified to make it perfectly clear when conclusions are based on PADDLE predictions versus experimental mutations of ADs. Several overreaching statements should be qualified regarding their conclusions that recruitment of Mediator through Med15 is the predominant interaction between ADs and cofactors driving transcriptional activation in vivo, and that recruitment of TFIID is secondary at best. The relevant locations in text are as follows:

– line 64 and 120: Given that the tiling approach in Figure 1 identified only 75-85% of previously reported ADs, these are likely to be overstatements.

– Figures 3F-3G, and Figure S7B should be labeled to make it clear that the ADs being analyzed were predicted; and the text in lines 215-219 should be revised to make it clear that the mutations were generated in silico and their effects on activation predicted by PADDLE.

– lines 218-219 and line 222: This is an overstatement as many of the conclusions are based on predictions by PADDLE, and might not stand up to experimental testing. The words "all" and "every" should be replaced by "many".

– 233-235: this is an overstatement as only 28 13-amino acid cADs were analyzed and only 19 of these appear to activate by the composition-driven mechanism.

– lines 250-252: Should be qualified to say that these features generally hold, as there are some grey areas in the results of Figure S4B, with some supposedly composition-driven cADs in which proline substitutions had significant effects, eg. War1 and Oaf3.

– lines 346-349: This seems like an overstatement as it is quite possible that other coactivators will contain subunits with ABD segments capable of the fuzzy interactions identified by Hahn and colleagues for the Med15 ABDs.

– lines 361-364: This is an overstatement, as there are certainly some TF tiles that bound TFIID and Med15 with similar affinity in Figure S4I-K. Also, these in vitro pulldown assays might not reflect recruitment in vivo or the extent to which TFIID vs. Mediator provides coactivator function at particular promoters.

– line 180-181: Was this true only for one AD, Arg81?

– lines 231-232: Indicate the correlation coefficient and associated p-value for the plot in Figure S4A.

– line 129: it would be more balanced to cite some of the key papers first showing the importance of hydrophobic amino acids in yeast transcriptional activators, including some or all of the following: PMID: 1846049, PMID: 9742254; PMID: 7862116; PMID: 8816468.

– lines 156-157 and 170-171: These two statements as written appear to be contradictory.

– line 311-312: Reference PMID: 19940160 seems relevant to this section.

– line 452: shouldn't Figure 6B be cited here also?

*Reviewer #2:*

Transcription activation domains (ADs) are poorly conserved and intrinsically disordered, but their description is based on a limited number of examples and a more general description of ADs and their binding characteristics has been lacking.

In the paper evaluated here, Sanborn et al. use an impressive variety of high throughput (HTP) methods and a novel deep learning algorithm to identify and characterize ADs. They initially tile 164 known yeast transcription factors to measure activation by 7460 protein segments, each 53 amino acids (aa), with a step size of 12-13 aa to measure their ability to activate transcription. This allows them to identify 150 ADs in 96 TAs, which are then characterized in different ways, again by HTP approaches, and used to train a deep learning computer model to predict the location and strength of acidic ADs in both yeast and humans. It does so with impressive precision. The authors also identify sequence and structural determinants for these domains, again using HTP. To more generally characterize the ability of these ADs to work with a general co-activator, the interaction with Mediator subunit 15 (Med15) is characterized, also by HTP methods. This results in the conclusion that Med15 is able to bind most ADs by making use of a shape-agnostic, fuzzy interface. Further mechanistic characterization results in a model for multivalent AD-Coactivator interaction, and for transcription activation in general.

The results are persuasive, and the conclusions supported by the findings. This is an important paper in the field.

*Reviewer #3:*

This paper addresses the longstanding question of the nature of transcription activation domains. The authors combine high-throughput biochemical identification and characterization of activation domains present in all known yeast transcription factors with sophisticated structural modeling using neural networks to generate a predictor (referred to as PADDLE) of activation domains that they demonstrate can be applied to higher eukaryotes. The conclusions of the paper are supported by the data.

As described below, I do have several suggestions for how the authors might expand on the description of their work to enable a better appreciation of their findings.

1) Using a combination of computational approaches and experimental validation of computational predictions, the authors have accomplished an impressive analysis of acidic activation domains. In so doing, they have defined common biochemical features of a large and important class of activation domains conserved from yeast to mammals and provide solutions to longstanding questions about activation domain structure and function. There are, however, transactivation domains that fall into other classes (e.g. glutamine-rich activation domains) and that may not have scored highly in their assay. It would be informative if the authors could comment on the relationship (or not) of their findings to other types of transactivation domains.

2) Since it is possible that a significant fraction of readers might not be specialists in computational modeling, it might be helpful if the authors could expand on their description of PADDLE. It would be interesting to know which features of PADDLE proved to be key to its predictive power (eg. percentage of acidic and bulky hydrophobic residues, secondary structure, disorder, other considerations). Which features of PADDLE make it superior to the authors' initial model that relied on activation domain composition of acidic and hydrophobic residues? Did using a convolutional neural network make a significant contribution to PADDLE and, if so, how?

3. Lines 135-138. The description of the relative abilities of Asp and Glu to promote activation and the results in Figure 2 figure supplement 1D is a little confusing to me. Would it be appropriate to say that Asp is more strongly associated with activation by wild type tiles than Glu, rather than Asp but not Glu is associated…? If not, the authors may wish to comment further on their observation that in a fair number of cases ADs with Asp residues mutated with Glu exhibit reduced but still substantial activation activity.

4. Line 311. The authors may wish to describe Med15 as the primary Mediator subunit targeted by "many ADs," since as shown in Figure 5C, while a remarkable 73% of ADs tested do bind MED15, there are still a significant number that don't.

5. Lines 386-388 and Figure 6, figure supplement 1B. It may be helpful to comment on the group of cAD domains whose activities are minimally affected (<3-fold) by PRO insertion but predicted by modeling to have a high percentage of residues in alpha helix when bound to KIX domain.

6. Line 991, legend to Figure 3, Supplement 1, panel A. I suspect the authors meant to refer to panel B, rather than C, for comparable PADDLE predictions.

7. Legend to Figure 6A. the legend refers to hydrophobic (yellow) and acidic (green) residues that form the AD-binding surfaces displayed. In the figure green residues are labeled as basic.

---

## [Author Response]

Essential revisions:Revise the manuscript text or figures to address the following comments:1. Text should be revised to eliminate overreaching statements claiming that all ADs or core ADs have been identified here, and that the principles governing the function of all ADs have been identified. Text and figures should also be modified to make it clear when conclusions are based on PADDLE predictions versus experimentally analyzed mutations of ADs. Several overreaching statements should be qualified regarding conclusions that recruitment of Mediator through Med15 is the predominant interaction between ADs and cofactors driving transcriptional activation in vivo, and that recruitment of TFIID is secondary at best. The relevant locations in text are as follows:– line 64 and 120: Given that the tiling approach in Figure 1 identified only 75-85% of previously reported ADs, these are likely to be overstatements.

We have removed claims of identifying “*all* ADs.”

– Figures 3F-3G, and Figure S7B should be labeled to make it clear that the ADs being analyzed were predicted; and the text in lines 215-219 should be revised to make it clear that the mutations were generated in silico and their effects on activation predicted by PADDLE.

We have clarified that analyses are based on “*predicted* activation.”

– lines 218-219 and line 222: This is an overstatement as many of the conclusions are based on predictions by PADDLE, and might not stand up to experimental testing. The words "all" and "every" should be replaced by "many".

We have made the requested changes.

– 233-235: this is an overstatement as only 28 13-amino acid cADs were analyzed and only 19 of these appear to activate by the composition-driven mechanism.

We have changed the sentence to refer to the “cADs *studied here*”.

– lines 250-252: Should be qualified to say that these features generally hold, as there are some grey areas in the results of Figure S4B, with some supposedly composition-driven cADs in which proline substitutions had significant effects, eg. War1 and Oaf3.

We have qualified these statements with “typically”.

– lines 346-349: This seems like an overstatement as it is quite possible that other coactivators will contain subunits with ABD segments capable of the fuzzy interactions identified by Hahn and colleagues for the Med15 ABDs.

The reviewer is correct that other coactivators contain ABDs that bind ADs in important ways. We feel that the statement referenced here is already appropriately qualified, since “Mediator recruitment is *a* key driver for gene activation and determined *largely* by AD binding to its Med15 subunit” does not rule out possibilities of other functional interactions.

– lines 361-364: This is an overstatement, as there are certainly some TF tiles that bound TFIID and Med15 with similar affinity in Figure S4I-K. Also, these in vitro pulldown assays might not reflect recruitment in vivo or the extent to which TFIID vs. Mediator provides coactivator function at particular promoters.

We have changed these sentences to refer to “*most* ADs” and “*most* TF-directed activation.”

2. Line 180-181: Was this true only for one AD, Arg81?

De novo PADDLE predictions were accurate across 22 TFs and the predictions for Arg81 in Figure 3B were just one example. We have updated the text to clarify this point.

Lines 231-232: Indicate the correlation coefficient and associated p-value for the plot in Figure S4A.

The Pearson’s r value (0.32) and P-value (0.09) have been added to the main text and the supplemental figure legend.

3. The definitive manner in which the authors state that they used 'all yeast TFs' (164) as their starting point requires explanation. Where do these come from? Who decided this was the complete list? The list of TFs used should be provided and its origin disclosed.

As described in the supplement, the TFs used in this study are all proteins labeled with Gene Ontology term “DNA-binding transcription factor activity” GO:003700. We have updated the main text to mention use of Gene Ontology terms in choosing proteins and added a list of all 164 proteins to Figure 1—Source Data 1.

4. At the end of page 5, line 101-102, the authors allude to cells being assay per tile and 99 being 'seen' in at least 10 cells. This may require better explanation for the uninitiated. At least this reviewer found it difficult to comprehend what was meant.

We have clarified this statement as follows: “Over 2,850,000 cells were assayed in total, giving excellent coverage of library elements (approximately 99% of TF tiles were observed in at least 10 cells each).”

5. In Figure 1D, please mark the positions of known (previously characterized) ADs.

Previously characterized ADs have been marked as black bars along the x-axis.

6. In 1F, a large proportion of the TFs do not contain an AD. This requires a comment or proposed explanation.

We have added the sentence “Consistent with the diverse functions of TFs beyond transcriptional activation (e.g. repression), 68 TFs did not contain any ADs (Figure 1F).”

7. The references to the Rosetta suite programs on page 26, line 373 is confusing. It should be moved to line 272 (after 'Rosetta Suite').

We have clarified this point. (We presume the reviewer meant line 372 instead of 272.)

8. Lines 135-138. The description of the relative abilities of Asp and Glu to promote activation and the results in Figure 2 figure supplement 1D is a little confusing to me. Would it be appropriate to say that Asp is more strongly associated with activation by wild type tiles than Glu, rather than saying that Asp but not Glu is associated…? If not, the authors should comment further on their observation that in a fair number of cases ADs with Asp residues mutated with Glu exhibit reduced but still substantial activation activity.

We have clarified the sentence to say “We also noticed that activation of wild-type tiles was *more strongly correlated with* the number of Asp residues than the number of Glu residues…”

9. Line 311. The authors should describe Med15 as the primary Mediator subunit targeted by "many ADs," rather than implying that it is the primary subunit targeted by all. As shown in Figure 5C, while a remarkable 73% of ADs tested do bind MED15, there are still a significant number that don't.

We have added the word “many,” as requested.

10. Lines 386-388 and Figure 6, figure supplement 1B. The authors should comment on the group of cAD domains whose activities are minimally affected (<3-fold) by PRO insertion but predicted by modeling to have a high percentage of residues in alpha helix when bound to KIX domain.

We have added the following sentence: “Some cADs that were minimally affected by proline insertion were modeled as alpha helical, possibly because disordered regions are underrepresented in protein structure databases on which Rosetta was trained.”

11. Line 991, legend to Figure 3, Supplement 1, panel A. I suspect the authors meant to refer to panel B, rather than C, for comparable PADDLE predictions.

Thank you, we have corrected this typo.

12. Legend to Figure 6A. the legend refers to hydrophobic (yellow) and acidic (green) residues that form the AD-binding surfaces displayed. In the figure green residues are labeled as basic.

Thank you, we have corrected this typo.

13. Consider revising the text to address the Recommendations to authors provided by each of the reviewers.

We have also addressed the reviewers’ additional questions and recommendations, as detailed below:

Reviewer #1:[…] – line 129: it would be more balanced to cite some of the key papers first showing the importance of hydrophobic amino acids in yeast transcriptional activators, including some or all of the following: PMID: 1846049, PMID: 9742254; PMID: 7862116; PMID: 8816468

We have added these citations.

– lines 156-157 and 170-171: These two statements as written appear to be contradictory

To clarify this, we have revised lines 156-157 to say that activation strength is determined “in large part*”* (rather than “primarily”) by acidic and hydrophobic content.

– line 311-312: Reference PMID: 19940160 seems relevant to this section.

We have added this citation.

– line 452: shouldn't Figure 6B be cited here also?

For clarity, the references to Figure 7 and supplements over multiple sentences have been consolidated into one place. (We presume the reviewer is referring to Figure 7B rather than 6B.)

Reviewer #3:[…] As described below, I do have several suggestions for how the authors might expand on the description of their work to enable a better appreciation of their findings.1) Using a combination of computational approaches and experimental validation of computational predictions, the authors have accomplished an impressive analysis of acidic activation domains. In so doing, they have defined common biochemical features of a large and important class of activation domains conserved from yeast to mammals and provide solutions to longstanding questions about activation domain structure and function. There are, however, transactivation domains that fall into other classes (e.g. glutamine-rich activation domains) and that may not have scored highly in their assay. It would be informative if the authors could comment on the relationship (or not) of their findings to other types of transactivation domains.

In lines 212-214, we have now noted that other classes of human ADs, such as glutamine-rich or proline-rich ADs, are not predictable by PADDLE since they are not present in *S. cerevisiae*.

2) Since it is possible that a significant fraction of readers might not be specialists in computational modeling, it might be helpful if the authors could expand on their description of PADDLE. It would be interesting to know which features of PADDLE proved to be key to its predictive power (eg. percentage of acidic and bulky hydrophobic residues, secondary structure, disorder, other considerations). Which features of PADDLE make it superior to the authors' initial model that relied on activation domain composition of acidic and hydrophobic residues? Did using a convolutional neural network make a significant contribution to PADDLE and, if so, how?

As noted in the main text, having access to the entire protein sequence allowed PADDLE to perform significantly better than a neural network based only on amino acid composition. Because the features important for PADDLE’s predicted power were not designed into the algorithm but rather learned from data, the individual features are not necessarily human interpretable. The analyses in Figure 3E-G, Figure 3—figure supplement 1H-I, and the experiments in Figure 4 and Figure 4—figure supplement 1 represent our attempts distill the key features of ADs from PADDLE and explore them using experiments.

Additionally, the supplemental methods section titled “Predicting activation from protein sequence” contains more details on the neural network training. For example, it notes that we also trained a version of PADDLE solely using the amino acid sequence, without inputting predicted secondary structure and predicted disorder, and the predictive power was only slightly reduced.